# piRNA loading triggers MIWI translocation from the intermitochondrial cement to chromatoid body during mouse spermatogenesis

Huan Wei [1,6], Jie Gao[2,6], Di-Hang Lin[3,6], Ruirong Geng[2,6], Jiaoyang Liao[3], Tian-Yu Huang[3], Guanyi Shang[2], Jiongjie Jing[2], Zong-Wei Fan[1], Duo Pan[3], Zi-Qi Yin[3], Tianming Li[2], Xinyu Liu[3], Shuang Zhao[3], Chen Chen [4], Jinsong Li [1,3,5], Xin Wang [1]✉, Deqiang Ding [2]✉ & Mo-Fang Liu [1,3,5]✉

The intermitochondrial cement (IMC) and chromatoid body (CB) are posited as central sites for piRNA activity in mice, with MIWI initially assembling in the IMC for piRNA processing before translocating to the CB for functional deployment. The regulatory mechanism underpinning MIWI translocation, however, has remained elusive. We unveil that piRNA loading is the trigger for MIWI translocation from the IMC to CB. Mechanistically, piRNA loading facilitates MIWI release from the IMC by weakening its ties with the mitochondria-anchored TDRKH. This, in turn, enables arginine methylation of MIWI, augmenting its binding affinity for TDRD6 and ensuring its integration within the CB. Notably, loss of piRNA-loading ability causes MIWI entrapment in the IMC and its destabilization in male germ cells, leading to defective spermatogenesis and male infertility in mice. Collectively, our findings establish the critical role of piRNA loading in MIWI translocation during spermatogenesis, offering new insights into piRNA biology in mammals.

Germ granules, the membrane-less organelles that are predominantly found in the cytoplasm of animal germ cells[1], are integral to various RNA-centric processes, including small RNA synthesis, mRNA storage, translation, and degradation[2–5]. While a plethora of germ granules with distinct compositions and functionalities have been identified in animal germ cells[3,5], the intermitochondrial cement (IMC) and chromatoid body (CB) stand out as the most extensively characterized in mammalian male germ cells[6]. The IMC is discernible as an electron-dense granule nestled among mitochondrial clusters, which first appears in fetal prospermatogonia, becomes pronounced in mid-pachytene spermatocytes, then disassembles gradually in concert with mitochondrial dispersion in late-pachytene spermatocytes, and vanishes entirely at the post-meiotic spermatid stage[6–8]. In contrast, CB precursors materialize in late-pachytene spermatocytes as diminutive

[1]Key Laboratory of Systems Health Science of Zhejiang Province, School of Life Science, Hangzhou Institute for Advanced Study, Hangzhou 310024; University of Chinese Academy of Sciences, Hangzhou, China. [2]Shanghai Key Laboratory of Maternal Fetal Medicine, Clinical and Translational Research Center, Shanghai First Maternity and Infant Hospital, Frontier Science Center for Stem Cell Research, School of Life Sciences and Technology, Tongji University, Shanghai 200092, China. [3]New Cornerstone Science Laboratory, State Key Laboratory of Molecular Biology, State Key Laboratory of Cell Biology, Shanghai Key Laboratory of Molecular Andrology, Shanghai Institute of Biochemistry and Cell Biology, Center for Excellence in Molecular Cell Science, Chinese Academy of Sciences; University of Chinese Academy of Sciences, Shanghai 200031, China. [4]Department of Animal Science, Michigan State University, East Lansing, Michigan 48824, USA. [5]School of Life Science and Technology, Shanghai Tech University, Shanghai 201210, China. [6]These authors contributed equally: Huan Wei, Jie Gao, Di-Hang Lin, Ruirong Geng. ✉e-mail: wx@ucas.ac.cn; dingdeqiang@tongji.edu.cn; mfliu@sibcb.ac.cn

granules localized near the nuclear envelope[8,9]. These precursors coalesce into the CB, a solitary, finely filamentous, lobulated, and perinuclear granule, in post-meiotic round spermatids[8,9]. It is noteworthy that the IMC and CB are postulated to serve as the platforms for piRNA processing and functionality in mammalian male germ cells, respectively[6]. This notion is substantiated by the enrichment of piRNA processing proteins such as mitochondria-anchored MitoPLD, GASZ, and TDRKH proteins in the IMC[10–12], and the presence of competent PIWI/piRNA complexes in the CB[13]. Despite the prevalent thought that the IMC furnishes precursor materials for the CB[6], the mechanistic underpinnings of how mature piRNAs produced in the IMC transition into functional entities within piRNA machinery in the CB remain elusive.

piRNAs are recognized to partner with proteins belonging to the PIWI clade of the Argonaute family, thereby facilitating their biological roles, which are essential for germline development in animals[14–18]. PIWI/piRNA complexes primarily function to silence transposable elements, thereby safeguarding the germline genome integrity[14–18]. Later studies, however, have demonstrated their roles in the regulation of protein-coding genes, expanding our understanding of PIWI/piRNA functions in animal germ cells[2,19–27]. In mice, the PIWI family proteins, MIWI, MILI, and MIWI2, are specifically expressed in male germ cells, which display temporal and spatial expression patterns during spermatogenesis[28–30]. In particular, MIWI expression initiates in pachytene spermatocytes and persists through post-meiotic spermatids[29,31]. Previous studies have shown that MIWI is initially recruited to the IMC through interaction with the mitochondrion-anchored Tudor domain-containing protein TDRKH for piRNA processing in mid-pachytene spermatocytes[32]. Thereafter, MIWI is enriched in the CB in round spermatids, where it, in complex with piRNAs, functions to regulate transposon and protein-coding mRNA transcripts[2,19,20,24,26,27,33]. Despite these insights, the mechanism underlying the sequential translocation of MIWI from the piRNA-producing IMC to the piRNA-functioning CB during male germ cell development has remained an enigmatic aspect of piRNA biology.

In this study, we elucidate the pivotal role of piRNA loading in mediating the translocation of MIWI from the IMC to CB. Our results indicate that piRNA loading instigates the disengagement of MIWI from TDRKH, facilitating its release from the IMC. Notably, this detachment primes the MIWI protein for arginine methylation, which subsequently fosters its association with TDRD6 via its methylated arginine residues, thereby culminating in its integration into the CB. By generating piRNA loading-deficient Miwi mutant mice, we demonstrate that loss of piRNA-loading ability in MIWI impedes its translocation from the IMC to CB during spermatogenesis. This obstruction critically undermines MIWI stability in developing male germ cells and leads to defective spermatogenesis and male infertility in mice. Collectively, our study sheds light on the molecular dynamics underpinning the sequential translocation of MIWI in developing male germ cells and establishes a mechanistic bridge between piRNA processing and functionality during male germ cell differentiation in mice.

## Results

### piRNA loading-deficient mutations in *Miwi* result in MIWI retention within IMC

Considering that MIWI is initially recruited to the IMC by TDRKH for piRNA processing in pachytene spermatocytes[32], and later localized to the CB in complex with piRNAs in round spermatids[6], we posited that piRNA loading could be instrumental in the translocation of MIWI from the IMC to CB during spermatogenesis in mice. To investigate this, we utilized CRISPR-Cas9 genome editing to generate mice with mutations in the PAZ (*Miwi*[Y346A/Y347A]) or MID-domains (*Miwi*[Y569L/K573E]) of *Miwi* (referred to as *Miwi*[YY/YY] and *Miwi*[YK/YK], respectively; Figs 1a and S1a). These mutants exhibit a deficiency in binding to the 3′ or 5′ end of piRNAs, as established in our study of MIWI mutant[34] and a study of

*Drosophila* PIWI mutants[35]. Potential off-target effects of the gRNAs on specific genomic regions were tentatively ruled out based on sequence similarity (Fig. S1b). By RNA co-immunoprecipitation (RIP), we found that the anti-MIWI antibody efficiently precipitated piRNAs in wild-type mouse testes but not in *Miwi*[YY/YY] and *Miwi*[YK/YK] mutant testes (Fig. 1b). This indicates that both the PAZ-domain and MID-domain mutations in MIWI significantly disrupt its piRNA-loading ability. As a control, we observed that MILI, another mouse PIWI protein, maintained a comparable piRNA-loading capacity in both wildtype and piRNA-loading-deficient *Miwi* mutant mice (Fig. S2). These findings confirm that both PAZ-domain and MID-domain MIWI mutants are deficient in piRNA loading in mouse testes.

Using the piRNA loading-deficient *Miwi* mutant mice, we investigated the impact of piRNA loading on the subcellular localization of MIWI in male germ cells. Through co-immunostaining of MIWI and the spermatocyte stage marker γH2AX, we observed that both wildtype and piRNA loading-deficient MIWI began to express in mid-pachytene spermatocytes (stages VII–VIII) and were similarly localized to the IMC (Fig. 1c). Consistent with previous observations[36,37], wildtype MIWI protein progressively disassociates from the IMC and integrates into the CB precursors in late-pachytene and diplotene spermatocytes (stages IX–XI), eventually localizing in the CB in post-meiotic round spermatids (Fig. 1c). In contrast, both MIWI[YY] and MIWI[YK] proteins consistently congregated into exceptionally large perinuclear polar granules instead of integrating into the CB precursors in late-pachytene and diplotene spermatocytes (Fig. 1c), reminiscent of the mitochondrial clustering observed in various piRNA biogenesis-deficient mutant mice[12,32,38,39]. By performing immunostaining of the mitochondrial outer membrane protein TDRKH, we verified the occurrence of mitochondrial clustering in *Miwi*[YY/YY] or *Miwi*[YK/YK] late-pachytene and diplotene spermatocytes (Fig. 1d). Intriguingly, such mitochondrial clustering was not observed in *Miwi*[−/−] spermatocytes (Fig. 1d), suggesting that the piRNA-loading-deficient MIWI mutants may exert a dominant-negative effect on mitochondrial clustering. These results together indicate that the loss of piRNA-loading ability leads to the retention of MIWI in the IMC, reinforcing the notion that piRNA loading is indispensable for the translocation of MIWI into the CB during the differentiation of male germ cells in mice.

To further substantiate these observations, we conducted co-immunostaining of MIWI and TDRKH proteins in mouse testes. We observed that MIWI and TDRKH were perfectly colocalized in mid-pachytene spermatocytes (stages VII–VIII) in the testes all of WT, *Miwi*[YY/YY], or *Miwi*[YK/YK] mice (Fig. 1e, left), indicating that piRNA loading-deficient mutations do not affect MIWI recruitment to the IMC. In late-pachytene and diplotene spermatocytes (stages IX–XI), wildtype MIWI dissociated from TDRKH and formed bright and large granules, whereas both MIWI[YY] and MIWI[YK] mutants remained colocalized with TDRKH (Fig. 1e, middle). This further verifies that piRNA loading-deficient mutations in MIWI lead to its retention in the IMC. We also observed that MILI and MVH, two other proteins that initially localize in the IMC and later in the CB during mouse spermatogenesis[37,40], successfully translocated from the IMC to the CB in both *Miwi*[YY/YY] and *Miwi*[YK/YK] testes (Fig. S3). This supports that the fundamental architecture of the IMC and CB is preserved in both *Miwi*[YY/YY] and *Miwi*[YK/YK] male germ cells. Taken together, these results indicate that piRNA loading-deficient mutations lead to a failure in MIWI translocation from the IMC to CB, suggesting that piRNA loading is a prerequisite for MIWI to be released from the IMC and subsequently integrated into the CB.

### piRNA loading promotes MIWI dissociation from TDRKH

We sought to investigate how piRNA loading regulates the translocation of MIWI from the IMC to CB. A previous study demonstrated that MIWI is recruited to the IMC through interaction with TDRKH[32]. Intriguingly, we observed through co-immunoprecipitation (co-IP) that the

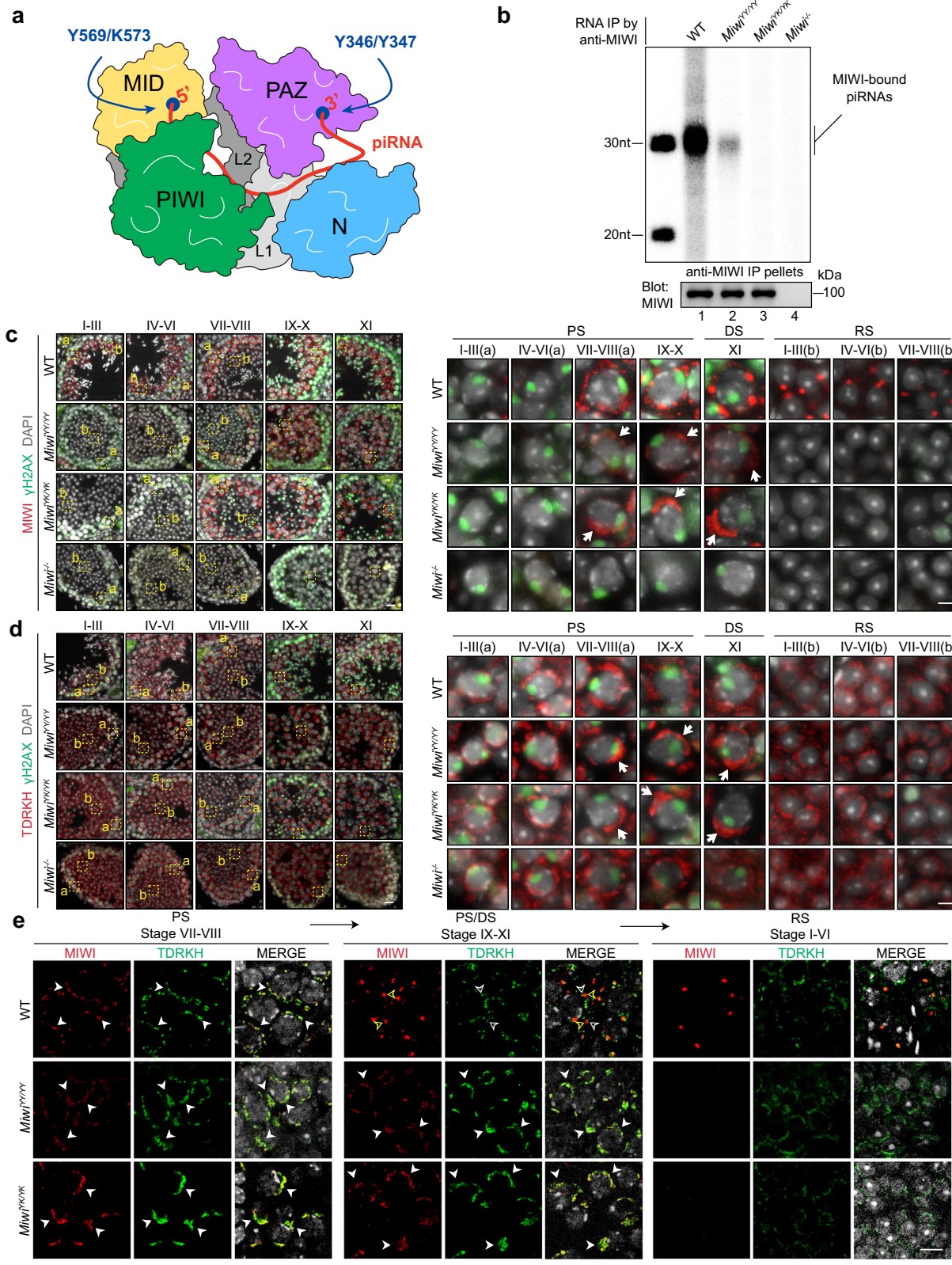

interaction between MIWI and TDRKH was markedly increased in the testes of 18 dpp (days post-partum) $Miwi^{YY/YY}$ and $Miwi^{YK/YK}$ mice compared to wildtype control (Fig. 2a, b). These findings led us to hypothesize that piRNA loading might facilitate the dissociation of MIWI from TDRKH. To test this hypothesis, we first examined whether TDRKH-associated MIWI protein is loaded with piRNAs in adult wildtype mouse testes. Using anti-MIWI immunoblotting as a reference for

loading, we discovered that MIWI protein in anti-TDRKH IP pellets was minimally associated with piRNAs compared to that in anti-MIWI IP control (Fig. 2c), suggesting that TDRKH predominantly interacts with piRNA-unloaded MIWI protein. Moreover, we serendipitously discovered that a commercially available rabbit monoclonal anti-MIWI antibody (ABclonal, Catalog No.: A3490, whose specificity was confirmed through immunoblotting of either epitope-deleted MIWI$^{\Delta aa701-801}$

**Fig. 1 | piRNA loading-deficient mutations cause the failure of MIWI releasing from the IMC in late-pachytene and diplotene spermatocytes. a** A schematic model showing the domain composition of MIWI and trajectory of the 5′ and 3′ ends of piRNA anchored with MID and PAZ-domain, respectively. The Y569/K573 and Y346/Y347 conserved in PIWI proteins are required for the 5′ end or 3′ end piRNA loading capacity of MIWI. **b** RNA co-IP assay of MIWI-associated-piRNAs (top) in wild-type (lane 1), *Miwi*[YY/YY] (lane 2), *Miwi*[YK/YK] (lane 3), and *Miwi*[−/−] testes (lane 4), with anti-MIWI IB as a loading reference (bottom). **c, d** Immunostaining of MIWI (**c**, red) and TDRKH (**d**, red) on testis sections from adult wildtype, *Miwi*[YY/YY], *Miwi*[YK/YK], and *Miwi*[−/−] mice using regular microscopy. Left: representative staining images of indicated mouse testis sections, scale bar, 20 μm; right, the enlargement of

yellow framed regions, scale bar, 5 μm. The developmental stages of spermatocytes and spermatids were distinguished according to γH2AX (green) and DAPI (grey-scale) staining. White arrows indicated MIWI (**c**) or TDRKH (**d**) aggregations. PS pachytene spermatocytes, DS diplotene spermatocytes, RS round spermatids. **e** Co-immunostaining of MIWI (red) and TDRKH (green) on testis sections from adult wildtype, *Miwi*[YY/YY] and *Miwi*[YK/YK] mice using confocal microscopy, with nuclei counterstained by DAPI (greyscale). White arrowheads indicated colocalization sites; yellow and white open arrowheads respectively indicated the unique localization of MIWI and TDRKH at non-colocalization sites. Scale bar, 10 μm. The results shown are representative of three independent experiments. Source data are provided as a Source Data file.

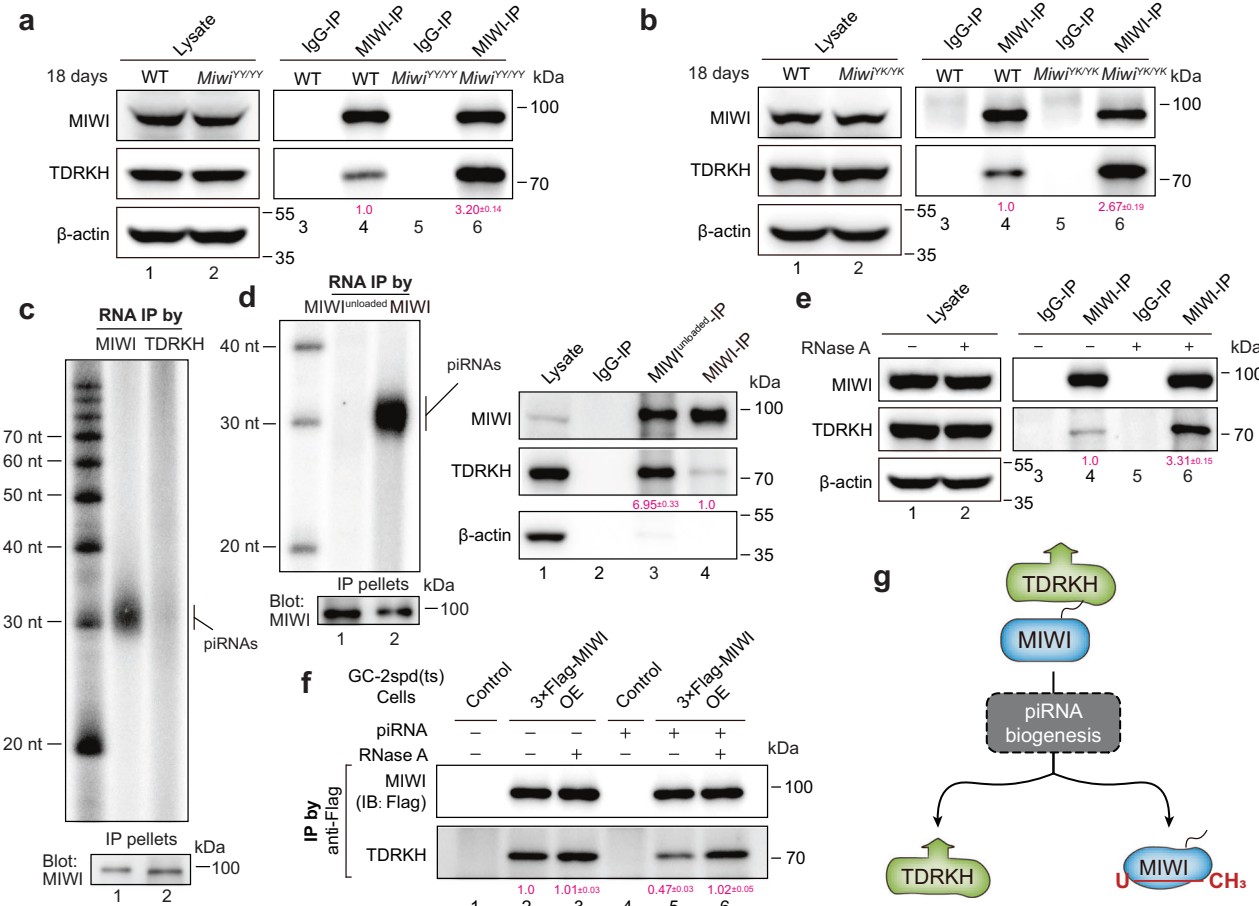

**Fig. 2 | piRNA loading facilitates MIWI dissociation with TDRKH. a, b** Co-IP assay of the association of MIWI and TDRKH in mouse testes from 18 dpp wildtype, *Miwi*[YY/YY], and *Miwi*[YK/YK] mice. Quantification of blot intensity of TDRKH protein in anti-MIWI pellets is shown in parentheses [the one from wildtype control mouse (lane 4) is set as 1.0 after normalization with MIWI blotting]. **c** RNA co-IP assay of MIWI-interacting piRNAs in anti-MIWI (lane 1) and anti-TDRKH IP pellets (lane 2) from adult wildtype mouse testes, with anti-MIWI IB as a loading reference (bottom). **d** Anti-MIWI[unloaded] preferably pulled down piRNA-unloaded MIWI (left) and TDRKH (right) in adult mouse testicular lysate. Left, RNA co-IP assays using anti-MIWI[unloaded] and control anti-MIWI antibodies in adult wildtype mouse testicular lysate, with anti-MIWI IB as loading references. Right, co-IP assay of the association of MIWI and TDRKH using anti-MIWI[unloaded] (lane 3) and control anti-MIWI antibodies (lane 4) in adult wildtype mouse testicular lysate, with testicular lysate (lane 1) and IgG IP (lane 2) serving as positive and negative controls, respectively. Quantification

of blot intensity of TDRKH is shown in parentheses [the one anti-MIWI IP (lane 4) is set as 1.0 after normalization with MIWI blotting]. **e** RNase A treatment enhanced the MIWI-TDRKH interaction in wild-type mouse testes. Quantification of blot intensity of TDRKH protein in anti-MIWI pellets is shown in parentheses [the one from RNase A-untreated (lane 4) is set as 1.0 after normalization with MIWI blotting]. **f** Transfection of piRNA attenuated the MIWI-TDRKH interaction in Flag-tagged MIWI-stable-expressed GC-2spd (ts) cells. OE overexpression. Quantification of blot intensity of TDRKH is shown in parentheses [the one with RNase A-untreated and piRNA-free condition in Flag-tagged MIWI-stable-expressed GC-2spd (ts) cell lysates (lane 2) is set as 1.0 after normalization with MIWI blotting]. **g** Schematic diagram showing that piRNA loading facilitates the dissociation of MIWI from TDRKH. The results shown are representative of three independent experiments. Quantification of western blot analysis are represented as mean ± SD. Source data are provided as a Source Data file.

mutant, or MIWI protein in wildtype or *Miwi*⁻/⁻ mouse testes; Fig. S4a) selectively immunoprecipitated piRNA-unloaded MIWI protein in mouse testes as well as in transfected HEK293T cells (Figs 2d and S4b). Henceforth, we referred to this antibody as anti-MIWI^unloaded. In sharp contrast, a rabbit polyclonal anti-MIWI antibody that we developed in-house[34] was able to immunoprecipitate both piRNA-loaded and unloaded MIWI protein. We reasoned that piRNA loading-induced protein conformational changes might occlude the epitope in MIWI recognized by the anti-MIWI^unloaded antibody. As expected, we found TDRKH to be much more enriched in anti-MIWI^unloaded IP compared to the anti-MIWI IP control (Fig. 2d, right). These results together indicate that TDRKH selectively interacts with piRNA-unloaded MIWI protein.

Consistent with our above observations, we found that the interaction between MIWI and TDRKH in adult wildtype testicular lysate was notably strengthened upon RNase A treatment (Fig. 2e), suggesting a repressive role of piRNA loading to the interaction of MIWI with TDRKH. To further corroborate this, we established a GC-2spd (ts) cell line stably expressing MIWI to directly assess the effect of piRNAs on the MIWI-TDRKH interaction (Fig. S4c). Indeed, piRNA transfection significantly attenuated the MIWI-TDRKH interaction (Fig. 2f, lane 2 vs lane 5), while treatment with RNase A effectively counteracted the repressive effect of piRNA on their interaction (lane 5 vs lane 6). Taken together, these findings suggest that piRNA loading triggers the disassociation of MIWI from the mitochondria-anchored TDRKH protein (Fig. 2g), thereby facilitating the translocation of MIWI from the IMC to CB during male germ cell development.

## piRNA loading promotes MIWI to interact with TDRD6 via enhancing its arginine methylation

Next, we sought to understand how MIWI is transported to the CB following its release from the IMC. Previous studies have demonstrated the necessity of TDRD6 for MIWI localization in the CB of mouse round spermatids[41]. Thus, we examined whether MIWI protein is recruited to TDRD6 following its release from the IMC. Through co-immunostaining, we observed that TDRD6 expression begins in mid-pachytene spermatocytes (Figs 3a and S5a, stages VII–VIII) with minimal colocalization with TDRKH, becomes highly abundant, and primarily enriches in distinct CB precursor granules in late-pachytene and diplotene spermatocytes (stages IX–XI), and ultimately localizes in the solitary CB in round spermatids. These observations suggest that the expression and localization of TDRD6 temporally and spatially coincide with MIWI translocation from the IMC to CB. Utilizing *Tdrd6* knockout mice that we generated via CRISPR-Cas9 genome editing (referred to as *Tdrd6*⁻/⁻; Fig. S5b, c), we discovered that *Tdrd6* depletion resulted in male infertility but had a negligible impact on piRNA biogenesis in mouse testes (Fig. S5d–h), consistent with previous findings[41]. We further confirmed that the levels of both MIWI protein and MIWI-associated piRNAs remained largely unaltered in *Tdrd6*⁻/⁻ testes (Fig. S5i, j), whereas MIWI condensation in the CB was significantly disrupted, if not entirely, in *Tdrd6*⁻/⁻ round spermatids (Fig. S6a). Intriguingly, double immunostaining of MIWI and TDRKH proteins revealed that MIWI localization was unaltered in *Tdrd6*⁻/⁻ mid-pachytene spermatocytes (stages VII–VIII; Fig. 3b, left), while in *Tdrd6*⁻/⁻ late-pachytene and diplotene spermatocytes (stages IX–XI) and round spermatids (middle and right), MIWI gradually dissociated from TDRKH despite a diffuse distribution. Consistently, our co-IP assay showed negligible alterations in the MIWI-TDRKH interaction in *Tdrd6*⁻/⁻ testes (Fig. S6b). These results together suggest that TDRD6 is not required for MIWI localization in the IMC or its release from the IMC, but crucial for the subsequent translocation of MIWI to the CB during the progression of mouse male germ cell differentiation.

Our above results demonstrated that piRNA loading leads to MIWI dissociation from TDRKH and its release from the IMC. We next asked whether piRNA loading also facilitates the TDRD6-mediated translocation of MIWI to the CB. Utilizing co-IP, we observed a marked reduction in the interaction between MIWI and TDRD6 in both *Miwi*^YY/YY and *Miwi*^YK/YK testes compared to wildtype control (Fig. 3c, d), suggesting that piRNA loading-deficient mutations in MIWI compromise its interaction with TDRD6. Supporting this, our co-immunostaining of testis sections revealed that the MIWI^YY and MIWI^YK proteins exhibited minimal overlap with the TDRD6 protein in late-pachytene and diplotene spermatocytes (stages IX–XI) and round spermatids, as opposed to the near-perfect colocalization observed between wildtype MIWI and TDRD6 (Fig. 3e). Interestingly, we observed that the absence of MIWI in CB precursors moderately altered the size of TDRD6 granules in late-pachytene and diplotene spermatocytes but little affected its enrichment in the CB in round spermatids (Figs 3e and S6c). These results indicate that piRNA loading is essential for MIWI protein to efficiently interact with TDRD6 in mouse male germ cells.

Despite a substantially reduced interaction between the piRNA loading-deficient mutant MIWI and TDRD6 (Fig. 3c–e), we unexpectedly found that depletion of piRNAs by RNase A treatment only marginally affected this interaction in wildtype mouse testicular lysate (Fig. 4a). This strongly suggests that piRNA loading might not directly mediate the MIWI-TDRD6 interaction. Considering that Tudor family proteins are thought to interact with PIWI proteins by binding to their methylated arginine residues in the N-termini[42,43], we next asked whether piRNA loading contributes to the MIWI-TDRD6 interaction by facilitating arginine methylation of MIWI. To address this, we first assessed whether the interaction of MIWI with TDRD6 depends on its arginine methylation. Utilizing a methylation-deficient MIWI mutant in which arginine (R) residues in all three RG motifs were substituted with lysine (K) residues at the N-terminus of MIWI (referred to as MIWI R-K) as previously reported[42], we observed that the loss of arginine methylation, as confirmed by SYM10 (Anti-dimethyl-arginine, symmetric) antibody immunoblotting, substantially weakened the MIWI-TDRD6 interaction in co-transfected cells (Fig. 4b). Moreover, treatment with MTA, a competitive inhibitor of methyltransferases, impaired the MIWI-TDRD6 interaction in a dosage-dependent manner (Fig. 4c). This strongly suggests that arginine methylation is necessary for MIWI to effectively interact with TDRD6.

We next investigated whether piRNA loading influences MIWI arginine methylation in testes. Our anti-SYM10 immunoblotting showed a significant decrease in MIWI arginine methylation in anti-MIWI IP samples from *Miwi*^YY/YY and *Miwi*^YK/YK testes compared to wildtype control (Fig. 4d, e), directly supporting the importance of piRNA loading for MIWI methylation. Additionally, by using anti-MIWI^unloaded and anti-MIWI antibodies to respectively immunoprecipitate piRNA-unloaded and loaded MIWI proteins from adult wildtype testes, we found a markedly lower level of arginine methylation in the anti-MIWI^unloaded IP samples relative to the anti-MIWI IP control (Fig. 4f), suggesting that piRNA-unloaded MIWI is less methylated than piRNA-loaded MIWI. Consistently, anti-MIWI^unloaded pulled down a much lower level of TDRD6 than anti-MIWI IP control (Fig. 4f), further showing a weak interaction of unmethylated MIWI with TDRD6 in mouse testes. These findings together indicate that piRNA loading augments the arginine methylation of MIWI, which in turn promotes its interaction with TDRD6 in mouse male germ cells.

We further aimed to elucidate the mechanism through which piRNA loading contributes to the arginine methylation of MIWI. A previous study shows that TDRKH interacts robustly with the N-terminus of MIWI in an arginine methylation-independent manner[44]. This led us to hypothesize that TDRKH might inhibit arginine methylation of MIWI by binding to its N-terminus. To assess this hypothesis, we respectively purified MIWI protein in anti-TDRKH and anti-MIWI IP samples from wild-type testes. Remarkably, we observed that MIWI protein isolated from an anti-TDRKH IP sample exhibited a substantially lower arginine methylation level compared to that from anti-MIWI IP control (Fig. 4g). This indicates that TDRKH preferentially

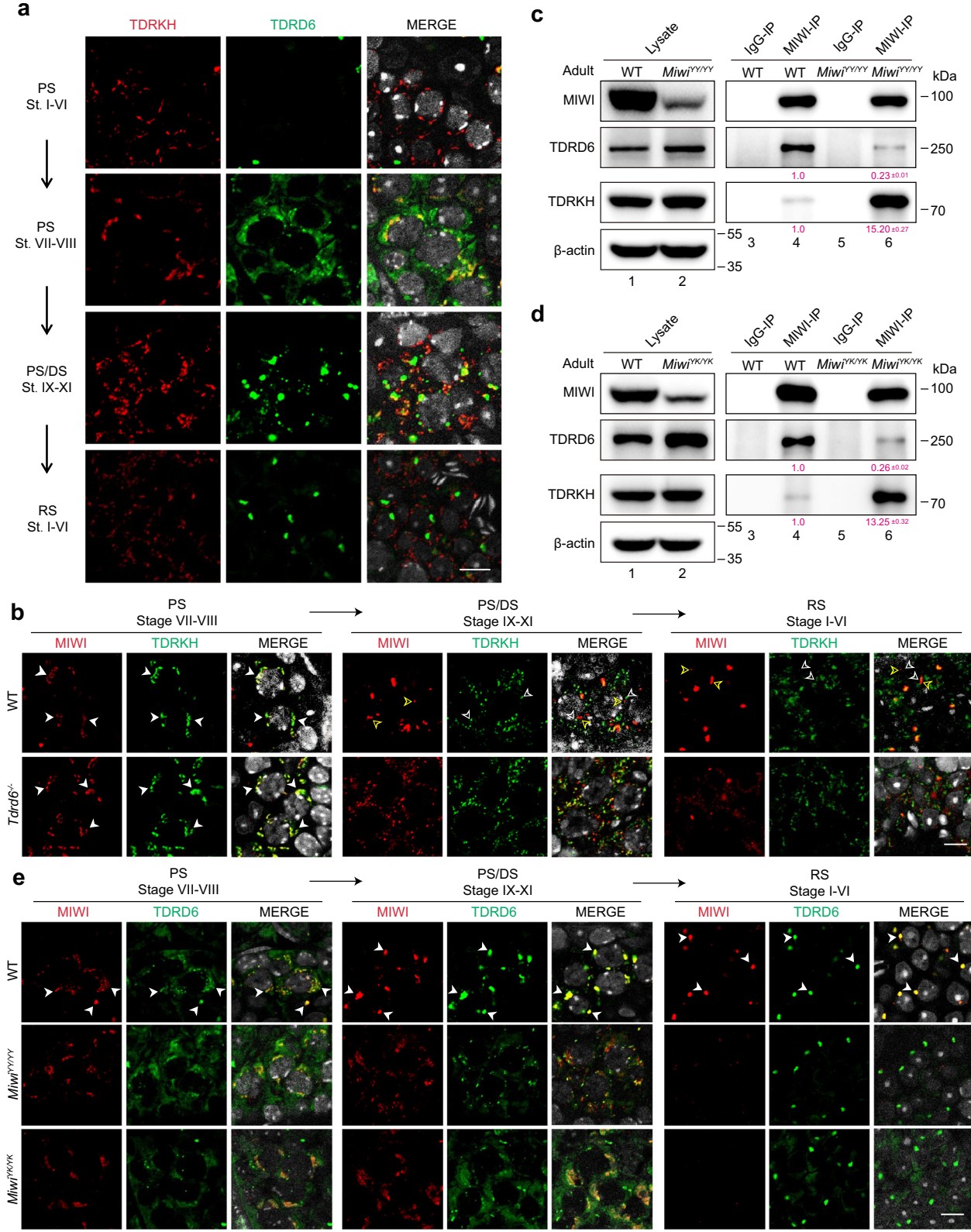

binds to unmodified MIWI in mouse testes, implying that TDRKH binding might occlude the arginine methylation of MIWI. Indeed, we found that TDRKH diminished the arginine methylation of MIWI in a dosage-dependent manner in co-transfected cells (Fig. 4h), directly supporting an inhibitory role of TDRKH in MIWI methylation. Taken together, these results indicate that piRNA loading promotes the dissociation of MIWI from TDRKH, subsequently facilitating MIWI arginine methylation to enhance its interaction with TDRD6 for integrating into the CB (Fig. 4i).

## piRNA loading-deficient mutations in *Miwi* lead to spermiogenic arrest in mice

Having demonstrated the functional necessity of piRNA loading in the translocation of MIWI from the IMC to CB during male germ cell

**Fig. 3 | piRNA loading is required for MIWI interaction with TDRD6 and transportation to the CB during male germ cell differentiation. a** Co-immunostaining of TDRKH (red) and TDRD6 (green) on testis sections from adult wildtype mice using confocal microscopy, with nuclei counterstained by DAPI (greyscale). PS pachytene spermatocytes, DS diplotene spermatocytes, RS round spermatids. Scale bar, 10 μm. **b** Co-immunostaining of MIWI (red) and TDRKH (green) on testis sections from adult wildtype and *Tdrd6⁻/⁻* mice using confocal microscopy, with nuclei counterstained by DAPI (greyscale). White arrowheads indicated colocalization sites; yellow and white open arrowheads respectively indicated the unique localization of MIWI and TDRKH at non-colocalization sites. Scale bar, 10 μm. **c**, **d** Co-IP assay of the association of MIWI with TDRD6 and TDRKH in mouse testes from adult wildtype (**c** and **d**, lanes 3 and 4), *Miwi^YY/YY*

(**c**, lanes 5 and 6) and *Miwi^YK/YK* (**d**, lanes 5 and 6) mice. Anti-MIWI IP pellets (**c** and **d**, lanes 4 and 6) were immunoblotted by the indicated antibodies, with testicular lysate (**c** and **d**, lanes 1 and 2) and IgG IP (**c** and **d**, lanes 3 and 5) serving as positive and negative controls, respectively. Quantification of blot intensity of indicated proteins in anti-MIWI IP pellets is shown in parentheses [the one from wildtype control mouse (lane 4) is set as 1.0 after normalization with MIWI blotting]. **e** Co-immunostaining of MIWI (red) and TDRD6 (green) on testis sections from adult wildtype, *Miwi^YY/YY* and *Miwi^YK/YK* mice using confocal microscopy, with nuclei counterstained by DAPI (greyscale). White arrowheads indicated colocalization sites. Scale bar, 10 μm. The results shown are representative of three independent experiments. Quantification of western blot analysis are represented as mean ± SD. Source data are provided as a Source Data file.

differentiation, we next investigated whether the piRNA-loading ability of MIWI is essential for spermatogenesis and male fertility in mice. When *Miwi^YY/YY* and *Miwi^YK/YK* male mice were crossbred with wildtype females, all observed homozygous *Miwi^YY/YY* and *Miwi^YK/YK* males were sterile, with notably smaller testes than their wildtype counterparts (Fig. 5a, b). Histological analysis complemented with acrosome staining of testicular sections using anti-ACRV1 or peanut agglutinin (PNA) revealed that spermatogenesis in both *Miwi^YY/YY* or *Miwi^YK/YK* testes was arrested at the round spermatid stage (Figs 5c, d and S6d). Moreover, we noted that the acrosome cap-containing round spermatids were much more detectable in *Miwi^YY/YY* and *Miwi^YK/YK* testes than that in *Miwi⁻/⁻* testes (Figs 5d and S6d), supporting that *Miwi^YY/YY* and *Miwi^YK/YK* spermatids could progress to a later stage compared with *Miwi⁻/⁻* controls. In line with this observation, apoptotic signals were prominently detected in the luminal layer of the seminiferous tubules in adult *Miwi^YY/YY*, *Miwi^YK/YK*, and *Miwi⁻/⁻* testes (Fig. 5e). These results together demonstrate that piRNA loading-deficient mutations in MIWI lead to spermiogenic failure and male infertility in mice.

### piRNA loading-deficient mutations in *Miwi* impair piRNA production and MIWI stability

We next investigated whether piRNA loading-deficient mutations in MIWI affect piRNA production. Through radiolabeling of total small RNAs, we observed a characteristic MIWI-bound piRNA population of ~30 nt in length in wildtype testes, but this piRNA population was absent in *Miwi^YY/YY*, *Miwi^YK/YK*, and *Miwi⁻/⁻* testes (Fig. 6a). This observation was corroborated by our small RNA-seq data, confirming a significant reduction in piRNAs of ~30 nt in length in *Miwi^YY/YY*, *Miwi^YK/YK*, and *Miwi⁻/⁻* testes (Fig. 6b). We found that piRNAs from both wildtype and *Miwi* mutant testes were derived from the same genomic loci and exhibited the expected 1U preference (Fig. 6c, d), but the abundance of piRNAs from each piRNA cluster was proportionally diminished in *Miwi^YY/YY*, *Miwi^YK/YK*, and *Miwi⁻/⁻* testes compared to wildtype controls (Fig. 6e). These results suggest that piRNA loading-deficient mutations in MIWI do not affect the selection of piRNA precursors but impair the production of the MIWI-associated piRNA population.

By immunostaining, we noted that both MIWI^YY and MIWI^YK proteins were easily detectable in spermatocytes, but their levels significantly decreased in round spermatids compared to wild-type controls (Fig. 1c, e). This suggests that piRNA-unloaded MIWI becomes unstable during male germ cell differentiation in mice. To substantiate this, we further analyzed MIWI expression in *Miwi^YY/YY* and *Miwi^YK/YK* testes at three developmental time points: 18 dpp encompassing pachytene spermatocytes, 20 dpp, which add diplotene spermatocytes and early round spermatids, and 24 dpp, which add step 4 round spermatids. We found that MIWI expression in *Miwi^YY/YY* and *Miwi^YK/YK* testes was similar to that in wildtype controls at 18 dpp but decreased at 20 dpp and even further at 24 dpp (Fig. 6f). Notably, MILI, another mouse PIWI protein, remained unaltered in *Miwi^YY/YY* and *Miwi^YK/YK* testes across all developmental stages (Fig. 6f). Our qPCR showed that *Miwi* mRNA level in mutant testes was unaltered compared to wildtype controls (Fig. S7a), suggesting that the decline in protein expression

could be attributed to the instability of piRNA-unloaded MIWI mutant proteins. In line with these results, we observed that RNase A treatment significantly increased the susceptibility of MIWI protein immuno-precipitated from wildtype mouse testes to thermolysin-mediated proteolysis, underscoring that piRNA-free MIWI is less stable than its piRNA-loaded counterpart (Fig. S7b, c). Consistently, we found a marked increase of MIWI ubiquitination in anti-MIWI IP pellets from *Miwi^YY/YY* and *Miwi^YK/YK* testes compared to wildtype control (Fig. S7d), suggesting that piRNA-loading-deficient MIWI proteins are subjected to degradation via the ubiquitin-proteasome pathway in mouse testes. Taken together, these results suggest that loss of piRNA-loading ability compromises not only the correct subcellular localization of MIWI but also significantly impairs its protein stability and piRNA production in mouse testes.

## Discussion

It is evident that MIWI is initially recruited to the IMC for piRNA production and then translocated to the CB for piRNA function during spermatogenesis, but the mechanisms underlying the MIWI translocation remain elusive. In this study, we demonstrate that piRNA loading is a prerequisite for MIWI to translocate from the IMC to CB during mouse male germ cell differentiation, shedding light on the molecular mechanisms governing the precise translocation of MIWI in developing male germ cells. As illustrated in Fig. 7, MIWI expression commences in mid-pachytene spermatocytes, where it is recruited to the IMC for piRNA processing through interaction with TDRKH. In the IMC, pre-piRNAs, generated by the endonuclease MitoPLD, load onto MIWI protein and then undergo 3′-end trimming by the 3′-to-5′-end exonuclease Trimmer/PNLDC1[38,45–48], following 2′-O-methylation of the 3′ end by Hen1/HENMT1 to produce mature piRNAs[49–53]. In late-pachytene and diplotene spermatocytes, MIWI becomes increasingly associated with piRNAs, leading to a diminished interaction with TDRKH. This change facilitates the release of MIWI from the IMC and exposes its N-terminal RG motifs for arginine methylation by PRMT5, which in turn enhances TDRD6 binding and primes MIWI integration in the CB. Collectively, our results delineate the pathway of MIWI translocation in developing male germ cells.

Furthermore, Tudor domains were initially identified as being capable of recognizing methylated arginine or lysine residues on target proteins and mediating protein-protein interactions[54]. Intriguingly, a previous study has shown that the Tudor domain of TDRKH recognizes and interacts with the N-terminal RG motif of MIWI, irrespective of arginine methylation[44], but the physiological significance of this particular characteristic of TDRKH remains unknown. Our results suggest that TDRKH acts as an initial receptor, interacting with unmethylated MIWI and simultaneously masking its methylation sites, and that the subsequent dissociation between them facilitates the arginine methylation of MIWI, which in turn strengthens its interaction with TDRD6 and promotes its subsequent translocation in the CB. Thus, the arginine methylation-modulated differential interaction of MIWI with the two Tudor family proteins TDRKH and TDRD6, dictates the localization of MIWI in specific germ granules during male germ

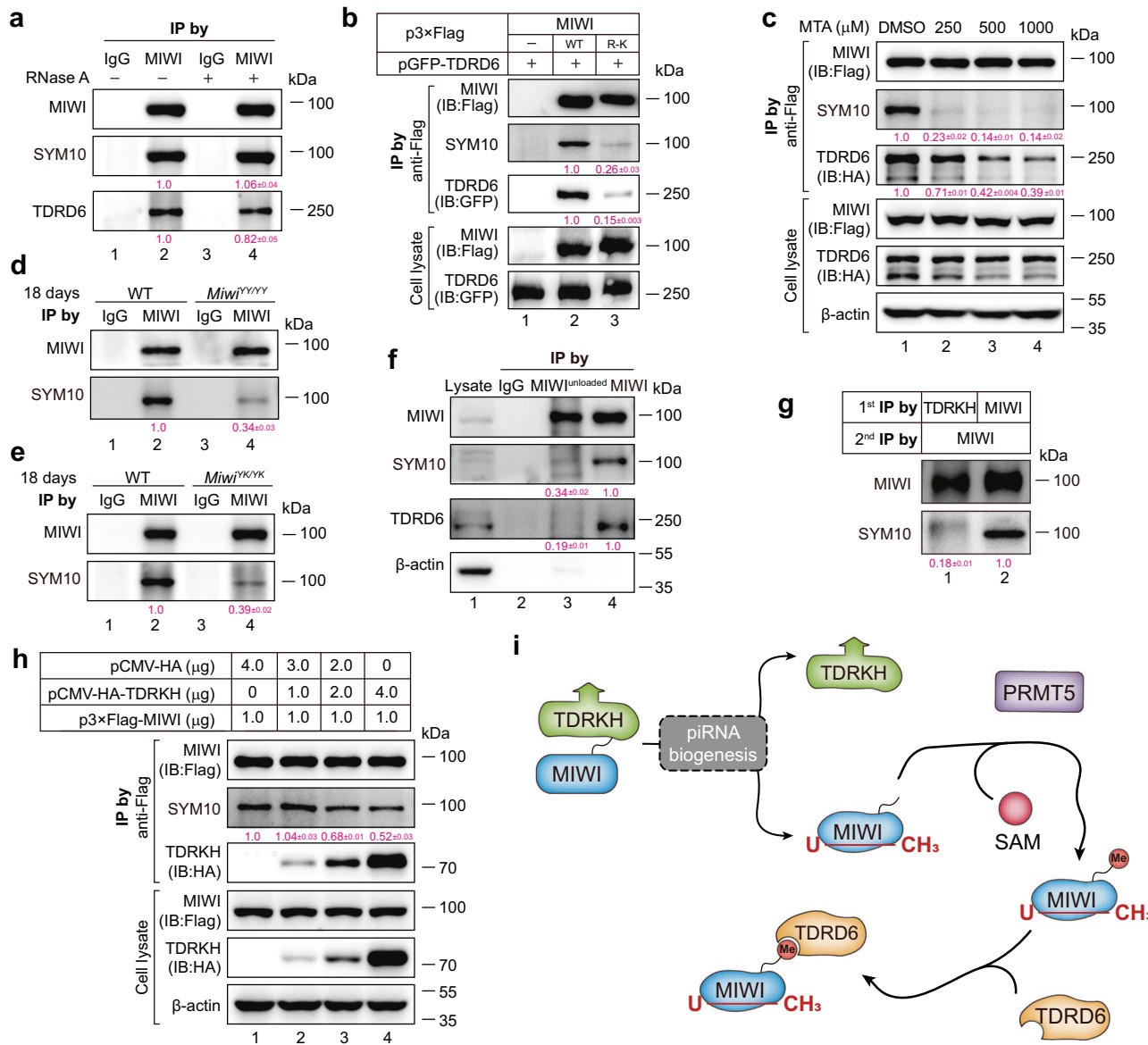

**Fig. 4 | piRNA loading enhances MIWI arginine methylation to facilitate its interaction with TDRD6. a** RNase A treatment barely altered MIWI methylation and MIWI-TDRD6 interaction in the adult testicular lysate. Quantification is shown in parentheses [the one from RNase A-untreated (lane 2) is set as 1.0 after normalization with MIWI blotting]. **b** Co-IP assay of the association of MIWI (lane 2) or arginine methylation-deficient MIWI[R-K] mutant (lane 3) with TDRD6 in co-transfected HEK293T cells. Quantification is shown in parentheses [the one with wildtype MIWI (lane 2) is set as 1.0 after normalization with MIWI blotting]. **c** Co-IP assay of the effect of methyltransferase inhibitor methylthioadenosine (MTA, Sigma, D5011) on the MIWI-TDRD6 interaction in co-transfected HEK293T cells. Quantification is shown in parentheses [the one with DMSO treatment (lane 1) is set as 1.0 after normalization with MIWI blotting]. **d, e** piRNA loading-deficient mutations impaired arginine methylation of MIWI in mouse testes. Quantification is shown in parentheses [the one from the wildtype control mouse (lane 2) is set as 1.0 after normalization with MIWI blotting]. **f** Anti-MIWI[unloaded] antibody pulled down

less methylated MIWI and TDRD6 in adult wildtype mouse testicular lysate (lane 3) compared with control anti-MIWI antibody (lane 4). Quantification is shown in parentheses [the one anti-MIWI IP (lane 4) is set as 1.0 after normalization with MIWI blotting]. **g** Sequential co-IP showing that TDRKH is mainly associated with unmethylated MIWI. Quantification is shown in parentheses [the first anti-MIWI IP (lane 2) is set as 1.0 after normalization with MIWI blotting]. **h** TDRKH reduced MIWI methylation in co-transfected HEK293T cells. Quantification is shown in parentheses [the one without TDRKH transfection (lane 1) is set as 1.0 after normalization with MIWI blotting]. **i** Schematic diagram showing that piRNA loading promotes MIWI dissociation from TDRKH, leading to the exposure of the N-terminal of MIWI for arginine methylation by PRMT5 to enhance the MIWI-TDRD6 interaction. The results shown are representative of three independent experiments. Quantification of western blot analysis are represented as mean ± SD. Source data are provided as a Source Data file.

cell differentiation, which underscores the multifaceted roles of Tudor family proteins in the piRNA pathway.

Interestingly, TDRKH is dispensable to the localization of MILI to mitochondria[32]. Moreover, despite that TDRKH depletion causes a dramatic reduction of MILI-associated piRNAs, MILI protein remains stable and normally localizes in the CB in *Tdrkh*-null round spermatids[32]. This suggests that piRNA loading is not required for

either the stability of MILI protein or its translocation from the IMC to CB. In fetal/neonatal prospermatogonia, MILI and MIWI2 are present to generate pre-pachytene piRNA[55], where they are respectively located in the IMC (also called pi-body), and piP-body and nucleus[56]. In particular, MIWI2 complexes are proposed to instantaneously associate with the IMC to participate in piRNA processing and then translocate to the piP-body and nucleus after being loaded with piRNAs[56]. Of note,

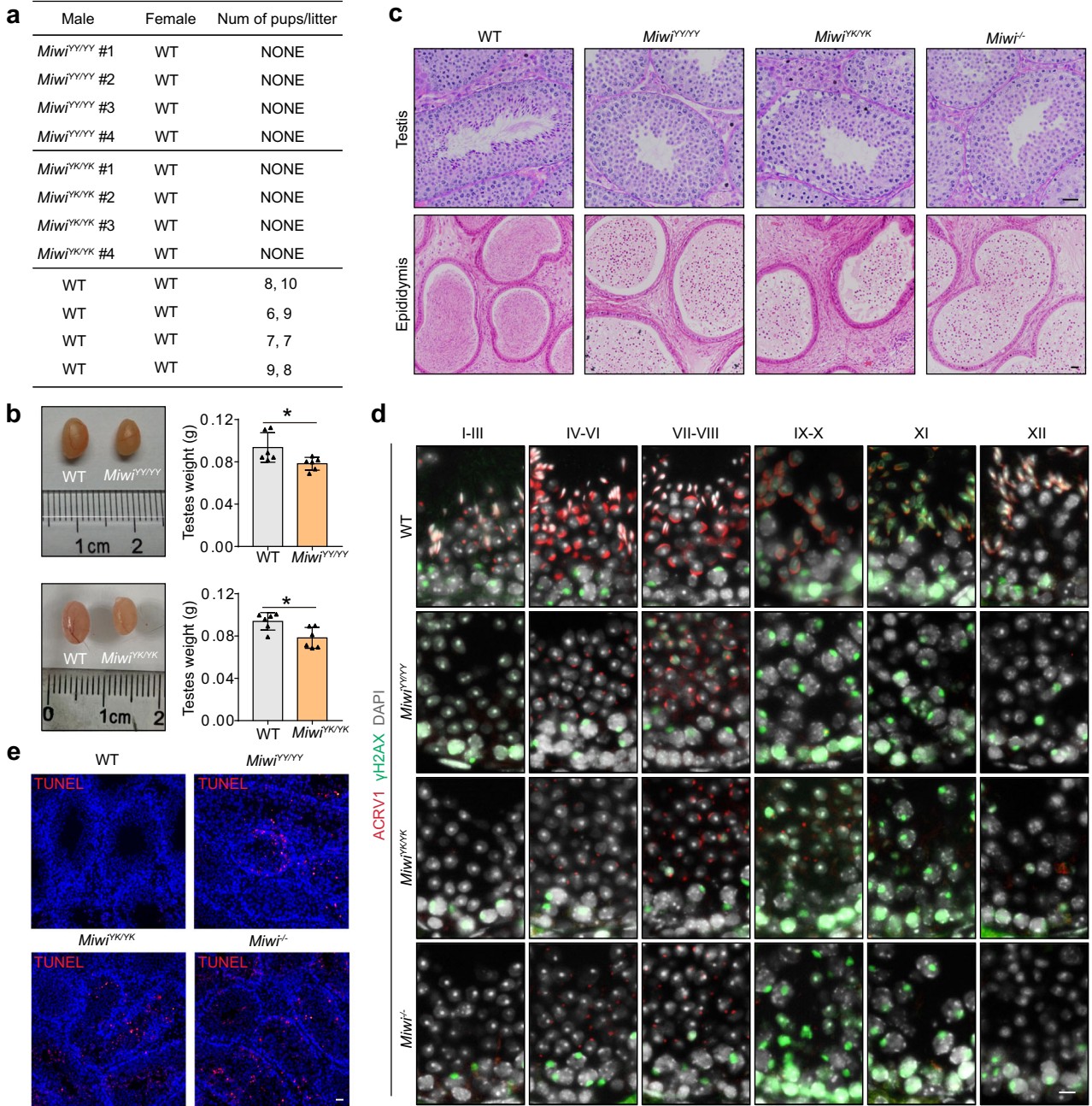

**Fig. 5 | piRNA loading-deficient mutations in *Miwi* lead to spermiogenic arrest in mice. a** All tested *Miwi*^YY/YY and *Miwi*^YK/YK males were infertile. **b** Testes from adult *Miwi*^YY/YY and *Miwi*^YK/YK mice were moderately reduced compared with wildtype control. Left, a representative image of testes from indicated mice; right, the average weight of testes from wildtype, *Miwi*^YY/YY ($p = 0.033$) and *Miwi*^YK/YK ($p = 0.014$) mice ($n = 6$, data are represented as mean ± SD, $P$ values were calculated using two-tailed Student's $t$-test, *$p < 0.05$). **c** PAS staining of the testis (top) and H&E staining of the epididymis (bottom) sections from adult wildtype, *Miwi*^YY/YY,

*Miwi*^YK/YK, and *Miwi*^−/− mice. Scale bar, 30 μm. **d** Acrosome staining (ACRV1, red) of testis sections from wildtype, *Miwi*^YY/YY, *Miwi*^YK/YK, and *Miwi*^−/− mice using regular microscopy. Developmental stages of the seminiferous tubules were distinguished according to γH2AX (green) and DAPI (grayscale) staining. Scale bar, 10 μm. **e** TUNEL assays (red) of testis sections from adult wildtype, *Miwi*^YY/YY, *Miwi*^YK/YK, and *Miwi*^−/− mice using regular microscopy, with nuclei counterstained by DAPI (blue). Scale bar, 30 μm. Results shown in **c**–**e** are representative of three independent experiments. Source data are provided as a Source Data file.

MIWI2 also interacts with TDRKH, despite it showing a lower affinity than MIWI[43]. This implies that TDRKH might play a role in MIWI2 recruitment to mitochondria. Additionally, TDRD9 specifically interacts with MIWI2, and they colocalize in the piP-body and nucleus[56,57], suggesting a role for TDRD9 in MIWI2 translocation akin to TDRD6 in MIWI translocation. Thus, our present study, combined with previous findings, suggests that mouse PIWI proteins employ distinct molecular mechanisms governing their translocation from the piRNA processing sites to functional deployment sites, but how MILI and MIWI2

translocation is precisely controlled requires further investigation in future studies.

Additionally, our current findings indicate that the MIWI protein is unstable in piRNA loading-deficient *Miwi* mutant mice. Consistent with this observation, MIWI protein was found to be unstable in several piRNA production-deficient mutant mice[32,38,58]. These findings together suggest that piRNA loading is critical for maintaining the stability of MIWI protein in developing male germ cells. It is plausible that piRNA loading directs the proper subcellular localization of MIWI in

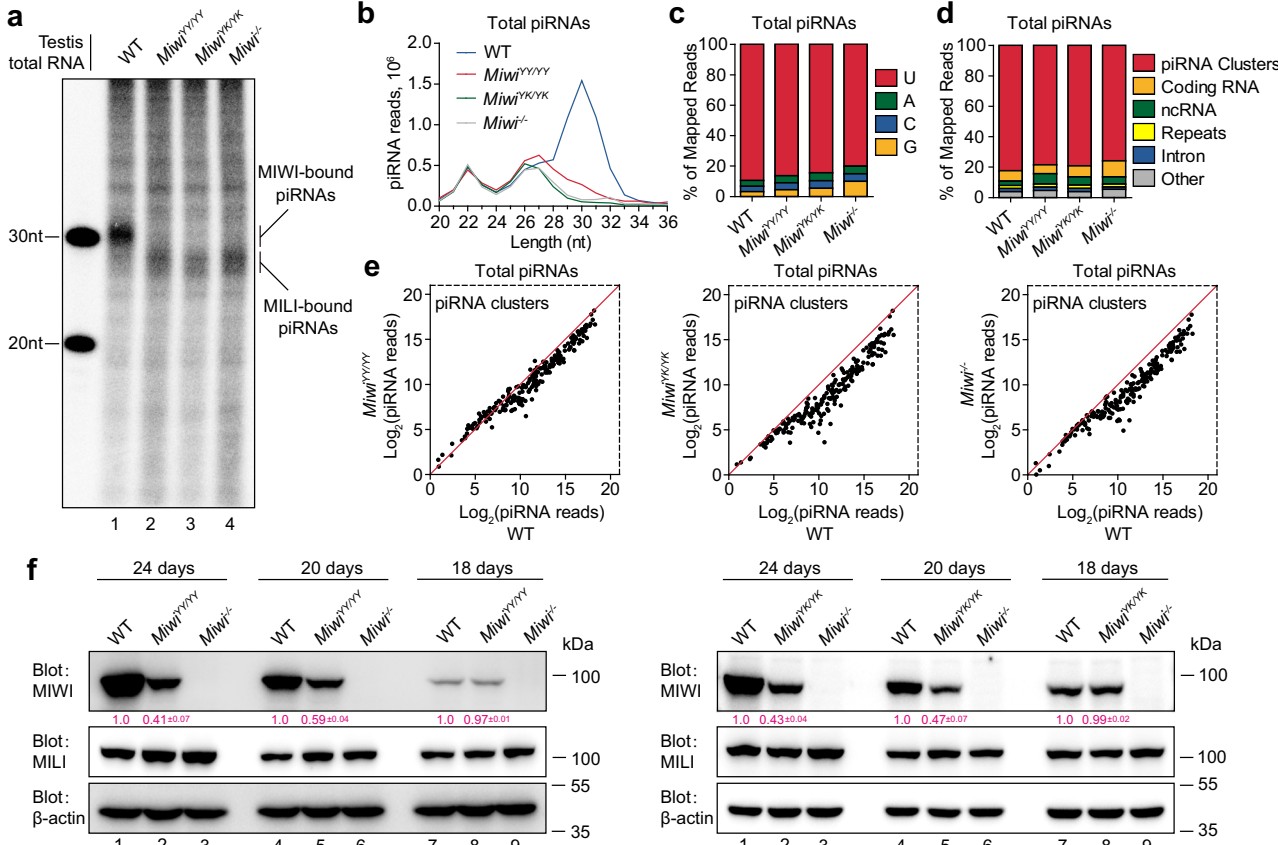

**Fig. 6 | piRNA loading-deficient mutations in *Miwi* impair piRNA production and MIWI stability in mouse testes. a** Detection of piRNA expression in adult wildtype, *Miwi*^YY/YY^, *Miwi*^YK/YK^, and *Miwi*^-/-^ testes. **b** The length distribution of small RNAs from adult wildtype, *Miwi*^YY/YY^, *Miwi*^YK/YK^, and *Miwi*^-/-^ testes. Data were normalized by miRNA reads (21–23 nt). **c** Nucleotide distributions at the first position in the piRNAs from adult wildtype, *Miwi*^YY/YY^, *Miwi*^YK/YK^, and *Miwi*^-/-^ testes. **d** Genomic annotation of the piRNAs from adult wildtype, *Miwi*^YY/YY^, *Miwi*^YK/YK^, and *Miwi*^-/-^ testes. The percentage of mapped reads is shown. **e** Scatter plot of total piRNA reads mapped to individual piRNA clusters from adult wildtype, *Miwi*^YY/YY^, *Miwi*^YK/YK^, and

*Miwi*^-/-^ testes. Data were normalized by miRNA reads (21–23 nt). **f** Western blotting of MIWI and MILI expression in testes from wildtype, *Miwi*^YY/YY^, *Miwi*^YK/YK^, and *Miwi*^-/-^ mice with indicated ages. β-actin served as a loading control. Quantification of blot intensity of MIWI is shown in parentheses (the one in wildtype testis is set as 1.0 after normalization with β-actin). Results shown in **a** and **f** are representative of three independent experiments, and small RNA-seq experiments shown in **b**–**e** with two replicates. Quantification of western blot analysis are represented as mean ± SD. Source data are provided as a Source Data file.

developing male germ cells, and this, in turn, protects MIWI from degradation. Interestingly, in a previous study of ours, we made the unexpected discovery that piRNAs trigger the ubiquitination of MIWI by the anaphase-promoting complex/cyclosome (APC/C) in late spermatids[34]. These findings together suggest that piRNA loading plays a dual role in regulating MIWI stability during spermatogenesis: it enhances the stability of piRNA-loaded MIWI for functions in the earlier developmental stages of male germ cells[59], but it also triggers the degradation and clearance of MIWI in late spermatids, thereby releasing the histone ubiquitin ligase RNF8, a MIWI-interacting protein, into the nucleus to mediate histone ubiquitination and subsequent histone-to-protamine exchange during late spermiogenesis[60].

In conclusion, our current study demonstrates that piRNA loading facilitates the translocation of MIWI from the IMC to CB during the progression of male germ cell differentiation in mice. This piRNA loading-regulated process only allows piRNA-loaded MIWI complexes transported to the CB, while "leftover" piRNA-unloaded MIWI proteins are degraded. This serves as a quality control mechanism, ensuring the delivery of competent MIWI/piRNA complexes to their functional site. After fulfilling their functions in the late stage of spermatid development, piRNA loading also triggers MIWI ubiquitination and degradation and promotes the coordinated elimination of MIWI and piRNAs, which is crucial for the histone-to-protamine exchange to produce functional sperm.

## Methods

### Ethics statement

All experimental animal procedures were approved by the Institutional Animal Care and Research Advisory Committee at SIBCB, CAS (2022-046). All experiments with mice were performed ethically according to the Guide for the Care and Use of Laboratory Animals and institutional guidelines.

### Cell culture and transfections

Female Human Embryonic Kidney 293T (HEK293T) cells (ATCC, CRL-3216) and male mouse spermatocyte-derived GC-2spd (ts) cells (ATCC, CRL-2196) were cultured in DMEM with 10% FBS according to the manufacturer's instructions. A GC-2spd (ts) cell line stably expressing MIWI was generated using the pLVX-Puro system (Clontech). Briefly, the coding sequence of MIWI was subcloned from pCMV-3×Flag-MIWI into the lentiviral vector pLVX-Puro. The lentiviral vectors were then co-transfected into HEK293T cells along with two packaging plasmids using Lipofectamine. After 48 h, the packaged pseudoviruses were harvested, filtered, and used to infect GC-2spd (ts) cells for 24 h. The infected GC-2spd (ts) cells were subsequently selected with 2 μg/mL puromycin. All cell lines were recently authenticated and tested for mycoplasma contamination. Transfection was performed with Lipofectamine 2000 (Thermo Fisher, 11668019) for HEK293T cells or Lipofectamine 3000 (Thermo Fisher, L3000015) for GC-2spd (ts) cells

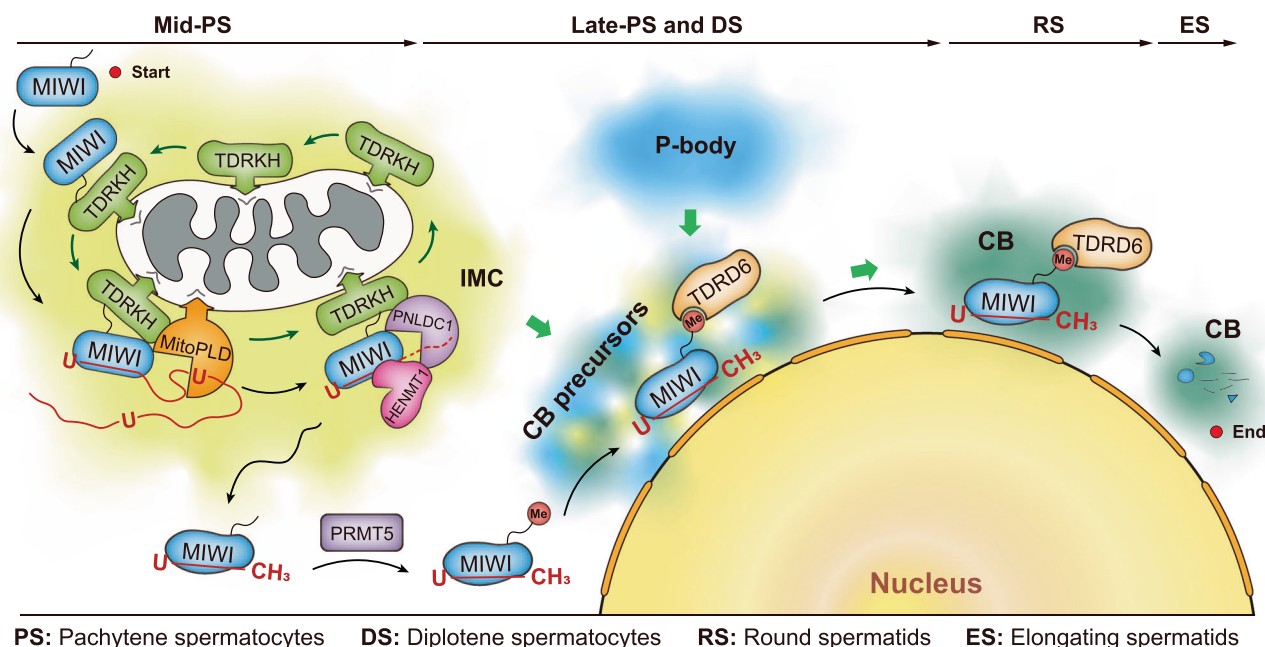

**PS:** Pachytene spermatocytes **DS:** Diplotene spermatocytes **RS:** Round spermatids **ES:** Elongating spermatids
**IMC:** Inter-mitochondrial cement **CB:** Chromatoid body **P-body:** Processing body

**Fig. 7 | Schematic model showing the panoramic view for the translocation of MIWI between germ granules during mouse spermatogenesis.** Upon its expression in mid-pachytene spermatocytes, MIWI protein is recruited to the IMC for piRNA processing via interacting with TDRKH through its unmethylated N-terminus, while piRNA loading induces a conformational change of MIWI and, in turn, weakens its interaction with TDRKH, leading to its release from the IMC. Meanwhile, the disassociation of MIWI with TDRKH simultaneously results in the arginine residues in its N-terminus exposed for methylation by PRMT5, thereby enhancing TDRD6 binding to prime its localization in the CB for piRNA function.

according to the manufacturer's instructions. All DNA plasmids were made free of endotoxins.

## Mice

Male C57BL/6 mice, including wildtype, *Miwi*^YY/YY^, *Miwi*^YK/YK^, *Miwi*^−/−^, and *Tdrd6*^−/−^ mutants, aged 18 dpp, 20 dpp, 24 dpp, or 6–8 weeks (adult), were used in this study. All mice were on the C57BL/6J genetic background. All mice were housed in the SIBCB animal facility under SPF conditions, under 12 h light/dark cycles in a pathogen-free room with clean bedding and free access to food and water, and temperature and humidity were kept at $23 \pm 1\,°C$, $55 \pm 5\%$. Cage and bedding changes were performed each week. *Miwi* knockout mice (*Miwi*^−/−^)[29] were purchased from Mutant Mouse Resource Research Centers (MMRRC). *Miwi*^YY/YY^ or *Miwi*^YK/YK^ knock-in mutant mice and *Tdrd6* knockout mice (*Tdrd6*^−/−^) were generated in the C57BL/6J background in this study, using a CRISPR/Cas9-mediated genome editing approach[61]. Briefly, Cas9 mRNA (100 ng/ml), sgRNA (50 ng/ml), and donor oligos (100 ng/ml) were mixed and injected into zygotes at the pronuclei stage. The injected zygotes were cultured in KSOM with amino acids at 37 °C under 5% $CO_2$ in the air until the blastocyst stage by 3.5 days. Thereafter, 15–25 blastocysts were transferred into the uteruses of pseudo-pregnant ICR females at 2.5 days post-coitum (dpc), and the offspring were subjected to genotyping. Potential off-target sites were selected according to CRISPR Design (http://zlab.bio/guide-design-resources) and examined by PCR-Sanger sequencing. Male mutant mice and littermate control mice were used in these analyses. The breeding studies were performed as described previously[62].

## Plasmids, oligonucleotides, and antibodies

pCMV-3×Flag-MIWI was constructed in our previous study[60]. Constructs pCMV-HA-TDRKH, pCMV-HA-TDRD6, and pGFP-TDRD6 were generated by inserting the cDNAs of mouse *Tdrkh* (NM_001357711.1) and *Tdrd6* (NM_001161366.1), excluding their 3′ UTRs, into pCMV-HA and pGFP, respectively. pCMV-3×Flag-MIWI^aa701-801^ and pCMV-3×Flag-

MIWI^R-K^ mutants were generated using KOD-Plus-mutagenesis kit (Toyobo, SMK-101). All constructs were verified through DNA sequencing. Oligonucleotides were synthesized by GenScript (Nanjing, China), and their sequences are provided in Supplementary Table 1. A list of all antibodies used is also included in Supplementary Table 1. In particular, the anti-MIWI^unloaded^ antibody (Catalog No.: A3490; see the details at https://abclonal.com.cn/catalog/A3490), which we found specifically recognizes piRNA-unloaded MIWI protein, was generated using a synthetic 100-amino acid (aa) peptide corresponding to position 700–800 of human PIWIL1 (Q96J94).

## Generation of TDRD6 antibody

Complementary DNA corresponding to TDRD6 1911–2135 aa (TDRD6, NP_001154838.1) was cloned into pET-28a (His-tag) vectors. His-tagged recombinant protein was used as the antigen to generate rabbit anti-TDRD6 polyclonal antisera. The antisera were affinity-purified with His-tagged TDRD6 antigen using an AminoLink Plus immobilization kit (Thermo Scientific).

## Immunoprecipitation and immunoblotting

Immunoprecipitation (IP) and immunoblotting (IB) assays were performed using standard IP and IB protocols. In brief, mouse testes or cells were homogenized in lysis buffer A [50 mM Tris-HCl (pH 7.4), 150 mM NaCl, 1% Triton X-100, 5 mM EDTA, and proteinase inhibitor cocktail (Roche, 4693132001)] or lysis buffer B [20 mM Tris-HCl (pH 7.0), 200 mM NaCl, 2.5 mM $MgCl_2$, 0.5% NP-40, 0.1% Triton X-100, and proteinase inhibitor cocktail (Roche, 4693132001)]. Primary antibody-coupled Protein A/G beads (Thermo Fisher, 88803) were added to the precleared tissue or cell lysates and incubated for 6 h at 4 °C. After washing with washing buffer [50 mM Tris-HCl (pH 7.4), 500 mM NaCl, 0.1% Triton X-100, 5 mM EDTA, and proteinase inhibitor cocktail], IP pellets or tissue/cell extracts were diluted in SDS-loading buffer and then analyzed with standard SDS-PAGE and IB procedures.

For sequential IP, the first IP pellets were eluted with 5 volumes of 0.1 M glycine-HCl (pH 3.0) and rotated at room temperature for 5 min. Tris buffer (pH 8.0) was added into the supernatant to adjust pH = 7.0, and the supernatant was incubated with the second antibody-coupled beads for 4 h at 4 °C. After washed with washing buffer, the second IP pellets were then analyzed with standard SDS-PAGE and IB procedures. Western blotting images were obtained by the Tanon-5200 Chemiluminescent Imaging System (Tanon).

For immunoprecipitation with RNase A Treatment, adult mouse testes or GC-2spd (ts) cells were homogenized in lysis buffer A [50 mM Tris-HCl (pH 7.4), 150 mM NaCl, 1% Triton X-100, 5 mM EDTA, proteinase inhibitor cocktail (Roche)] and treated with 250 µg/mL RNase A (Thermo Fisher, EN0531), followed by incubation for 1 h at 4 °C. Additional 125 µg/mL RNase A (Thermo Fisher) and primary antibody-coupled Protein A/G beads were added to the precleared tissue or cell lysates and incubated for 5 h at 4 °C to fully degrade RNAs. After washing with washing buffer [50 mM Tris-HCl (pH 7.4), 500 mM NaCl, 0.1% Triton X-100, 5 mM EDTA, and proteinase inhibitor cocktail], IP pellets or tissue/cell extracts were diluted in SDS-loading buffer and then analyzed with standard SDS-PAGE and IB procedures.

### Immunofluorescence

Testes were fixed overnight at 4 °C in PBS containing 4% PFA and then embedded in paraffin. Tissue sections were cut to a thickness of 5 µm, dewaxed, and rehydrated. Antigen retrieval was achieved by microwaving the sections in 0.01 M Tris-EDTA buffer (pH 9.0) for 2 min. After rinsing with PBS, the sections were blocked with 5% normal goat serum (NGS) for 30 min. The sections were subsequently incubated with primary antibodies diluted in 5% NGS at 37 °C for 1 h. The antibodies used included: anti-ACRV1 (1:50; 14040-1-AP, Proteintech), anti-MIWI (1:100; 2079, Cell Signaling Technology), anti-TDRKH (1:200; 13528-1-AP, Proteintech), anti-MILI (1:100; PM044, MBL), anti-MVH (1:200; ab13840, Abcam), anti-TDRD6 (1:200; homemade), or FITC-conjugated anti-γH2AX (1:500; 16–202 A, Millipore). PNA (Peanut agglutinin, Invitrogen, L32460) conjugated with Alexa Fluor Cy3 were used for acrosome staining. After washing with PBS, the sections were incubated with Alexa Fluor 555 anti-rabbit IgG (1:500; A31572, Thermo Fisher) and mounted using an antifade mounting medium containing DAPI (Beyotime, P0131). Fluorescence microscopy was carried out using a CKX53 fluorescence microscope (Olympus, Japan) or a Ti2-E confocal microscope (Nikon, Japan). Image J software (National Institutes of Health, Bethesda, MD, USA) was employed for grayscale conversion of DAPI signals.

For co-immunostaining of MIWI with TDRKH, TDRD6 with TDRKH, and TDRD6 with MIWI, tissue sections were first incubated with either anti-MIWI (1:100; 2079, Cell Signaling Technology) or anti-TDRKH (1:200; 13528-1-AP, Proteintech), followed by Alexa Fluor 555 anti-rabbit IgG. The sections were then re-blocked with 5% NGS and incubated with either anti-TDRKH or anti-TDRD6, which were pre-labeled with the Zenon Alexa Fluor 488 Rabbit IgG Labeling Kit (Thermo Fisher, Z25302) according to the manufacturer's instructions.

### RT-qPCR, RNA co-IP (RIP) assays

For RT-qPCR, total RNAs were extracted from indicated tissues using RNAiso Plus (Takara, 9109) as per the manufacturer's guidelines. Following the removal of residual genomic DNA with Turbo DNase (Invitrogen), 500 ng of total RNA was reverse-transcribed into cDNAs using the PrimeScript RT Reagent Kit (Takara, RR037A). RT-qPCR was performed on a QuantStudio 3 instrument (Thermo Fisher) utilizing a SYBR Premix Ex Taq kit (Takara, RR820A). Relative gene expression was analyzed based on the $2^{-\Delta\Delta Ct}$ method with β-actin as an internal control. For RIP assay of MIWI- or MILI-associated piRNAs, total RNAs were extracted from anti-MIWI or -MILI IP pellets and then labeled with [γ-$^{32}$P]-ATP (PerkinElmer) by T4 PNK (Thermo Fisher, EK0032) after

FastAP (Thermo Fisher, EF0651) treatment. They were resolved on a 15%, 7 M urea polyacrylamide gel along with radioactive oligoribonucleotide size markers, followed by autoradiography. All primers used for RT-qPCR are listed in Supplementary Table 1.

### In vitro piRNA loading assay

In vitro piRNA loading assay was performed as we recently described[63]. In brief, HEK293T cells were transfected with the Flag-tagged MIWI expression vector using Lipofectamine 2000 (Invitrogen). After 24 h, cells were harvested and lysed using lysis buffer [20 mM HEPES–KOH (pH 7.0), 1.5 mM MgCl$_2$, 100 mM NaCl, 0.1 mM EDTA, 0.1 mM DTT, and 10% v/v glycerol, 1% triton X-100 and 1×Complete EDTA-free protease inhibitor tablets (Roche)]. For loading MIWI with synthetic piRNA, precleared lysates were incubated with Cy5-labeled synthetic piRNA for 1 h at 37 °C. Subsequently, lysates were incubated with Protein A/G beads (Invitrogen) coupled with anti-MIWI or anti-MIWI$^{unloaded}$ antibodies, under continuous rotation for 4 h at 4 °C. After stringent washing, half of the anti-MIWI or anti-MIWI$^{unloaded}$ IPed beads were used for immunoblotting of MIWI, while the remaining half were used to determine the fluorescence intensities of Cy5.

### Periodic acid Schiff staining, hematoxylin and eosin staining, and apoptosis assays

For Periodic Acid Schiff (PAS, Solarbio, G1281) staining or Hematoxylin and Eosin (H&E) staining, testicular or epididymal tissues were fixed in Bouin's buffer, embedded in paraffin, and sectioned at 5 µm thickness. The paraffin-embedded sections were then sequentially deparaffinized, rehydrated, and stained with either Hematoxylin and Eosin or Periodic Acid Schiff. Apoptosis assays were performed using the In Situ Cell Death Detection Kit, TMR red (Roche, 12156792910), according to the manufacturer's protocols.

### Small RNA libraries and bioinformatic analyses

Small RNA libraries from immunoprecipitated RNAs or total RNA were prepared using the NEBNext® Multiplex Small RNA Library Prep Kit (NEB, E7300S), following the manufacturer's instructions. Libraries with different barcodes were then pooled together and sequenced using the Illumina NovaSeq 6000 (Novogene Co., Ltd.).

Sequenced reads were processed using Cutadapt to trim the sequencing adapters. The trimmed reads were then filtered based on length (24–32 nt) and aligned to several sets of sequences: piRNA clusters[31], coding RNAs, non-coding RNAs, repeats, and introns. Reads that did not map to any of the five sets of sequences were categorized as "other". The alignments were performed with Bowtie, allowing for one base mismatch. The category "repeats" included classes of repeats as defined by RepeatMasker (ftp://hgdownload.cse.ucsc.edu/goldenPath/mm10/database/rmsk.txt.gz). For total piRNA analyses, RNA reads were normalized by miRNA counts (21–23 nt). For analyses of MILI-bound piRNAs, RNA reads were normalized by the total reads in each library.

### Quantification of western blot analysis

Image J software (National Institutes of Health, Bethesda, MD, USA) was employed for the quantification of western blot analysis. The signal from each band was converted into intensity values. These values were subsequently normalized and utilized to calculate fold changes, providing a basis for comparing protein expression across various samples. The samples in the same panel were derived from the same experiment, and the gels/blots were processed in parallel. Results are presented as mean ± standard deviation (SD) of three separate experiments.

### Statistics

The numbers (*n*) of biological replicates or animals used are indicated in the individual figure legends. The experiments were not

randomized, and no statistical methods were employed to pre-determine the sample size. We utilized Student's *t*-test to compare the differences between treated groups and their respective paired controls. All statistical tests were two-sided. Results are presented as the mean ± standard deviation (SD). *P* values are indicated either in the text or on the figures, with values <0.05 (denoted by asterisks) considered significant (\*\*\**P* < 0.001, \*\**P* < 0.01).

## Reporting summary
Further information on research design is available in the Nature Portfolio Reporting Summary linked to this article.

## Data availability
The data supporting the findings of this study are available from the corresponding authors upon request. The small RNA-seq data used in this study are available in the BioProject field of the National Center for Biotechnology Information (NCBI) under accession code PRJNA979781. The RepeatMasker [ftp://hgdownload.cse.ucsc.edu/goldenPath/mm10/database/rmsk.txt.gz] was used to define the category "repeats". Source data are provided with this paper.

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

## Acknowledgements

We thank members of the Mo-Fang Liu' lab and Deqiang Ding' lab for their helpful comments. This work was supported by grants from the National Key R&D Program of China (No. 2022YFA1303300 to M.-F.L., No. 2022YFA1305300 to D.D., and No. 2021YFC2700200 to X.W.), National Natural Science Foundation of China (No. 32330053 to M.-F.L., No. 32271347 to X.W., No. 32070841 to D.D., No. 31830109 to M.-F.L., No. 31821004 to M.-F.L., No. 32101037 to X.W., and No. 32071286 to S.Z.), Science and Technology Commission of Shanghai Municipality (No. 23JC1403801 to M.-F.L., No. 20ZR1460000 to D.D., No. 21YF1452700 to X.W., and No. 21ZR1470500 to X.W.), the Young Elite Scientist Sponsorship Program of the China Association for Science and Technology (No. 2021QNRC001 to X.W.), Zhejiang Provincial Natural Science Foundation of China (No. LZ23C050001 to X.W.), the Research Funds of Hangzhou Institute for Advanced Study, UCAS(No. 2023HIAS-Y022 to X.W.), the Fundamental Research Funds for the Central Universities (No. 22120230292 to D.D.), the Foundation of Key Laboratory of Gene Engineering of the Ministry of Education, and the New Cornerstone Science Foundation (No. NCI202329 to M.-F.L.). Chen Chen was supported by NIH grants (R01HD084494 and R01GM132490).

## Author contributions

M.-F.L., D.D., and X.W. planned the project. H.W., J.G., D.-H.L., R.G., C.C., J.L., X.W., D.D., and M.-F.L. designed the experiments. H.W., J.G., D.-H.L., J.L., T.-Y.H., G.S., J.J., Z.-W.F., D.P., Z.-Q.Y., T.L., X.L., and S.Z. conducted the experiments. H.W., J.G., D.-H.L., X.W., and D.D. analyzed the data. R.G. was responsible for bioinformatic analyses. H.W., J.G., C.C., X.W., D.D., and M.-F.L. wrote the paper. All authors discussed the results and commented on the manuscript. M.-F.L., D.D., and X.W. supervised the study.

## Competing interests

The authors declare no competing interests.
