## [Peer Review File · Nature Communications]

piRNA loading triggers MIWI translocation from the intermitochondrial cement to chromatoid body during mouse spermatogenesisREVIEWER COMMENTS

Reviewer #1 (Remarks to the Author):

This interesting study investigates the link between loading of germline small RNAs (piRNAs) into their binding partners called PIWI proteins, and their localization in distinct cytoplasm granular structures. Using knock-in mice expressing point mutant versions of the PIWI protein MIWI that are unable to be loaded with piRNAs, they show its presence in the intermitochondrial cement (IMC). While the wildtype MIWI that is loaded with piRNA moves through the IMC to accumulate in the chromatoid body (CB). These distributions are dictated by interaction with distinct Tudor domain containing proteins (TDRDs): TDRKH in IMC and TDRD6 in CB. The proposal is that loading with piRNAs in the IMC promotes arginine methylation of MIWI such that it promotes interaction with TDRD6 and recruitment into CB. The authors summarize the data in a model that suggests a dynamic process of piRNA biogenesis in the IMC and function in CB, all of which is regulated by interaction with Tudor proteins and arginine methylation.

Comments:

1. It is confusing to label an antibody as anti-MIWIA3490. What else does this antibody stain in the lysate (Fig. S4)? For first-time characterization of a special antibody that recognizes unloaded MIWI, a full gel picture after immunoblotting should be shown. Did the authors map the epitope(s) of this antibody? Is it polyclonal, maybe also show somewhere what antigen was used to make it.
2. Line 149-150: is it true that the in-house antibody pulls down only loaded MIWI? Fig. 2a and b shows that it also pulls down MIWI mutant that is not able to be loaded. Please rephrase this sentence.
3. Fig. 4i. In this model there is no mention that TDRKH and recruitment of Trimmer for maturing piRNA 3' ends. The authors should consider discussing it. It is strange to show a mature piRNA being loaded into MIWI, as piRNAs mature once pre-piRNAs are loaded into MIWI.
4. Line 383: "washing" not "washed".

Reviewer #2 (Remarks to the Author):

The manuscript by Wei et al., shows, to the best of my knowledge, for the first time the functional link between the IMC as a place of piRNA generation and the CB as a place of piRNA activity. This link has been generally inferred but never substantiated. The authors show that piRNA loading of Miwi leads to arginine methylation which results in its release from Tdrkh, component of the IMC, and subsequent binding to Tdrd6, component of the CB.

The manuscript is generally well written and the presented data convincing. It takes our knowledge of the piRNA pathway a step further by elucidating in detail a mechanism of MIWI-piRNA translocation. By doing so this work contributes significantly to our understanding of piRNA-pathway biology.

Some comments/suggestions are respectfully made.

1. The manuscript contains many images showing immunofluorescent data. Unfortunately the authors chose to depict DAPI in dark blue. In many cases this makes the nucleus very hard to see. Claims that IF signal is "perinuclear" are now sometimes difficult to verify. Also, the morphology of the nucleus of meiotic cells can be used to determine at which meiotic stage they are. This is now in most instances not possible. Could the authors show all DAPI images in greyscale? This would make the data much easier to appreciate. See for an example: PMID 21383078.
2. The authors present data that piRNA loading results in Miwi methylation and subsequent binding to Tdrd6. The co-localization of Tdrd6 to the IMC is used as an argument that the association from Miwi with Tdrkh and Tdrd6 is a 2-step process. However, the level of magnification, the presentation of the images and the reagents used could be improved on. 1) I would like to see higher magnification

images of the IMC and Tdrd6 localization, 2) showing each of these stainings in greyscale and an overlap of the stainings in color. The Zenon kit is used to work around the problem of staining simultaneously with 2 antibodies derived from the same species (used for colocalization claims Fig. 1e and 3a, b, e). This tool generally works well but some level of cross-staining is always there. 3) I would like to see Tdrd6 staining combined with a non-rabbit antibody against the IMC or mitochondria (since the latter two are in such close proximity).

3. In general it would help the reader if instances of co-localization or absence thereof are indicated with arrows in the IF panels.

4. Also, could the authors indicate in the manuscript which images were obtained by confocal microscopy and which one by regular microscopy? Especially for the claims of co-localization this is important.

5. An important tool is the use of an antibody, A3490, which recognizes Miwi only when it's not associated with a piRNA. Figure S4 shows the validation of this antibody. Could the authors expand this figure by providing details on the epitope it was raised against? Also an immunofluorescence staining with this antibody on WT mouse testis material combined with 1) a marker against IMC/Mitochondria and, separately, 2) the CB (without using the Zenon kit) would be very insightful.

6. Figure 3e PS/DS The TDRD6 panel suggests that loss of loading ability MIWI affects TDRD6 granules (larger granules are absent in the mutant panels). More likely the stages of these meiotic cells are not similar. Can the authors provide images of spermatocytes in a more similar stage (or alternatively expand on a necessity for MIWI loading for correct Tdrd6 localization to pre-cursor CBs)?

7. Figure 3e RS, MIWI staining. Can the authors provide these images with extended overexposure? Right now I cannot determine whether signal is absent or radically reduced.

8. Figure 5c+d. The authors claim that the spermiogenic arrest in YY/YY and YK/YK mice is later than in Miwi KO. This is a very interesting observation. However, I cannot verify this with these images. Can the authors add additional data or more detailed images that underwrites their observation? Also, can the authors exclude that this difference is due to differences in genetic background of these mice (which can have a profound background on the timing of arrest in piRNA mutants)?

9. In the discussion there is some redundancy from line 310 on. Also, I'm curious to read the authors thoughts on Mili and the difference with Miwi in the piRNAs they generate/associate with in the light of their data. Finally, in the pull down experiment described by Vagin et al (G&D), Tdrd2/Tdrkh was indicated as an interactor of MIWI2, which could suggest that a similar mechanism is occurring in fetal germ cells too (though not with Tdrd6 since it is not expressed in these cells). What are the authors thoughts on this?

10. sentence 294 contains a linguistic error.

11. Sentence 297, "MIWI migration" maybe "MIWI translocation"?

Reviewer #3 (Remarks to the Author):

Wie and colleagues set up to test if piRNA loading would impact MIWI translocation from the intermitochondrial cement (IMC), a primary piRNA processing sites to the chromatoid body (CB), a functional deployment site inside germ cells. They make use of fine, well executed mouse genetics, biochemistry, and microscopy to show that loading is important for dissociation from TDRKH (a previously known interaction and a marker of the IMC) and later association with TDRD6 (a previously known interaction and a marker of the CB). They show that arginine methylation is also somewhat dependent on loading (a previously known fact, however needed to be tested in their mutant experimental setup), and that loading is key to proper spermiogenesis (also a previously known fact, however needed to be tested in their mutant experimental setup).

Overall, the experimental system and the experiments are very neat and well executed and should be published upon revision of the current manuscript. They do in general allow the conclusions drawn in the manuscript despite not adding a lot to the field in terms of novelty. My main argument for the above statement is that, as a mechanism for dissociation of MIWI from TDRKH, "piRNA loading" is

slightly beyond current standards in the field. MIWI mutants for piRNA loading and their effect in localization and spermiogenesis have been known for about a decade. It is a high bar to ask within the context of manuscript revisions, but the field would really benefit from understanding which allosteric changes occur upon piRNA loading and how they impact key protein-protein interactions. That would be an entire new structural biology manuscript. It does however make a point about the impact of the work presented here.

I present below two main and one minor points for improving the current manuscript

Role of TDRKH in occluding arginine methylation remains unclear. Is it because it masks methylation sites or is it because the same conformational changes happening upon piRNA loading exposes them? Would the same dose-dependent effect seen for TDRKH be observed for presence of piRNAs? This could be tested in the ectopic HEK cell system with/without piRNA transfections in presence/absence of the protein partners and using the A3490 antibody.

The same corresponding author has published in the past (PMID: 23328397) data showing the importance of piRNA loading for ubiquitination and Miwi degradation. In here, mutants that fail to properly load piRNA are shown to be destabilized as proteins. Considering the contradiction, I believe the authors ought to test whether there is increased or decreased ubiquitination in piRNA loading mutants and whether the same APC/C-ubiquitination pathway is at play or if another pathway is working upon clearance of MIWI that failed on loading piRNAs.

3) In addition, providing all uncropped original western blots would be ideal given the importance and extent of the biochemistry work in the manuscript. That helps other researchers better gauge and plan related future experiments.

Below is the point-by-point response to comments from the 3 reviewers.

Reviewer 1 (Remarks to the Author):

This interesting study investigates the link between loading of germline small RNAs (piRNAs) into their binding partners called PIWI proteins, and their localization in distinct cytoplasm granular structures.

Using knock-in mice expressing point mutant versions of the PIWI protein MIWI that are unable to be loaded with piRNAs, they show its presence in the intermitochondrial cement (IMC). While the wildtype MIWI that is loaded with piRNA moves through the IMC to accumulate in the chromatoid body (CB). These distributions are dictated by interaction with distinct Tudor domain containing proteins (TDRDs): TDRKH in IMC and TDRD6 in CB. The proposal is that loading with piRNAs in the IMC promotes arginine methylation of MIWI such that it promotes interaction with TDRD6 and recruitment into CB. The authors summarize the data in a model that suggests a dynamic process of piRNA biogenesis in the IMC and function in CB, all of which is regulated by interaction with Tudor proteins and arginine methylation.

This reviewer has concisely summarized all key discoveries presented in our manuscript. We thank the Reviewer for his/her appraisal of our work as an “interesting study”. We also greatly appreciate his/her constructive criticisms for us to further improve the manuscript, as detailed below.

Comments:

1. It is confusing to label an antibody as anti-MIWI^{A3490}. What else does this antibody stain in the lysate (Fig. S4)? For first-time characterization of a special antibody that recognizes unloaded MIWI, a full gel picture after immunoblotting should be shown. Did the authors map the epitope(s) of this antibody? Is it polyclonal, maybe also show somewhere what antigen was used to make it.

We apologize for missing this information in our previous version of manuscript. We have now provided a detailed description of this antibody in the **Plasmids, oligonucleotides, and antibodies** section of revised “Materials and Methods” (Lines 397-401). Actually, this antibody was manufactured by ABclonal Technology Co., Ltd. (Catalog No.: A3490; see the details at <https://abclonal.com.cn/catalog/A3490>), which was generated using a synthetic 100 amino acid (aa) peptide corresponding to position 700-800 of human PIWIL1 (HIWI, Q96J94). To avoid potential confusion, we have

now referred to this antibody as anti-MIWI^{unloaded} in the revised text.

In our previous manuscript, we confirmed the specificity of this antibody through immunoblotting of MIWI protein in wildtype or *Miwi*^{-/-} mouse testes. To further verify its specificity, we have now generated an epitope-deleted MIWI mutant by deletion of 100-aa epitope sequence within MIWI protein (referred to as MIWI^{Δaa701-801}). Using anti-Flag antibody as a positive control, anti-MIWI^{unloaded} antibody could effectively detect Flag-tagged wild-type MIWI but not MIWI^{Δaa701-801} mutant in transfected HEK293T cells (revised Fig. S4a, bottom; compare lanes 1 to 2), directly supporting the specificity of this antibody. Per Reviewer's suggestion, we have included the full immunoblotting gel images in revised Fig. S4a. We have described these results in the revised text (Lines 145-147).

2. Line 149-150: is it true that the in-house antibody pulls down only loaded MIWI? Fig. 2a and b shows that it also pulls down MIWI mutant that is not able to be loaded. Please rephrase this sentence.

Actually, our previous study has shown that this in-house antibody is able to immunoprecipitate both piRNA-loaded and unloaded MIWI protein (Zhao et al., *Dev Cell* **24**, 13-25, 2013). To avoid potential confusion, we have now rephrased this sentence as "In sharp contrast, a rabbit polyclonal anti-MIWI antibody that we developed in-house was able to immunoprecipitate both piRNA-loaded and unloaded MIWI protein" (Lines 148-151).

3. Fig. 4i. In this model there is no mention that TDRKH and recruitment of Trimmer for maturing piRNA 3' ends. The authors should consider discussing it. It is strange to show a mature piRNA being loaded into MIWI, as piRNAs mature once pre-piRNAs are loaded into MIWI.

We thank the Reviewer for his/her insightful comments. Per Reviewer's suggestion, we have now added a step of "piRNA biogenesis" into the models presented in revised Fig. 2g and 4i and incorporated the 3' end maturation process in our working model in revised Fig. 7. In brief, pre-piRNAs, produced by the endonuclease MitoPLD, undergo 3'-end trimming by the 3'-to-5'-end exonuclease Trimmer/PNLDC1 upon loading onto MIWI protein (Ding et al., *Nat Commun* **8**, 819, 2017; Izumi et al., *Cell* **164**, 962-973, 2016; Nishimura et al., *EMBO Rep* **19**, 2018; Tang et al., *Cell* **164**, 974-984, 2016; Y. Zhang et al., *Cell Res* **27**, 1392-1396, 2017), following 2'-O-methylation of the 3' end by Hen1/HENMT1 to produce mature piRNAs (Horwich et al., *Curr Biol* **17**, 1265-

1272, 2007; Kirino & Mourelatos, *Nat Struct Mol Biol* **14**, 347-348, 2007; Lim et al., *PLoS Genet* **11**, e1005620, 2015; Ohara et al., *Nat Struct Mol Biol* **14**, 349-350, 2007; Saito et al., *Genes Dev* **21**, 1603-1608, 2007). We have discussed this point in the revised text (Lines 301-303).

4. Line 383: "washing" not "washed".

We thank the Reviewer for pointing out this error and have corrected it in the revised text (Line 413).

Reviewer 2 (Remarks to the Author):

The manuscript by Wei et al., shows, to the best of my knowledge, for the first time the functional link between the IMC as a place of piRNA generation and the CB as a place of piRNA activity. This link has been generally inferred but never substantiated. The authors show that piRNA loading of Miwi leads to arginine methylation which results in its release from Tdrkh, component of the IMC, and subsequent binding to Tdrd6, component of the CB.

The manuscript is generally well written and the presented data convincing. It takes our knowledge of the piRNA pathway a step further by elucidating in detail a mechanism of MIWI-piRNA translocation. By doing so this work contributes significantly to our understanding of piRNA-pathway biology.

We greatly appreciate the positive appraisal of the Reviewer on our work. We have thus put major efforts in addressing his/her remaining concerns to make the story better, as detailed below.

Some comments/suggestions are respectfully made.

1. The manuscript contains many images showing immunofluorescent data. Unfortunately, the authors chose to depict DAPI in dark blue. In many cases this makes the nucleus very hard to see. Claims that IF signal is "perinuclear" are now sometimes difficult to verify. Also, the morphology of the nucleus of meiotic cells can be used to determine at which meiotic stage they are. This is now in most instances not possible. Could the authors show all DAPI images in greyscale? This would make the data much easier to appreciate. See for an example: PMID 21383078.

We thank the Reviewer for this suggestion. Accordingly, we have presented the DAPI staining in greyscale in revised Fig. 1, Fig. 3, Fig. 5, Fig. S3, Fig. S5 and Fig. S6, which indeed makes the data much easier to appreciate.

2. The authors present data that piRNA loading results in Miwi methylation and subsequent binding to Tdrd6. The co-localization of Tdrd6 to the IMC is used as an argument that the association from Miwi with Tdrkh and Tdrd6 is a 2-step process. However, the level of magnification, the presentation of the images and the reagents used could be improved on. 1) I would like to see higher magnification images of the IMC and Tdrd6 localization, 2) showing each of these stainings in greyscale and an overlap of the stainings in color. The Zenon kit is used to work around the problem of staining simultaneously with 2 antibodies derived from the same species (used for colocalization claims Fig. 1e and 3a, b, e). This tool generally works well but some level of cross-staining is always there. 3) I would like to see Tdrd6 staining combined with a non-rabbit antibody against the IMC or mitochondria (since the latter two are in such close proximity).

We thank the Reviewer for these suggestions. To better exhibit the localization of TDRKH and TDRD6 proteins in developing male germ cells, we have presented the images with a higher magnification in revised Fig. 3a. We have also attempted to present TDRKH or TDRD6 staining in grayscale with their respective staining overlaid in color, but we found it difficult to differentiate the two proteins (appended Fig. R1). Hence, we have now shown TDRKH staining in red, TDRD6 staining in green, and DAPI staining in grayscale in revised Fig. 3a. Moreover, we have now displayed DAPI staining in greyscale in all related figure panels.

To exclude potential cross-staining with Zenon Kit labeling in co-localization assay, we have co-stained the mitochondrial marker Cytochrome c using a widely recognized mouse monoclonal antibody (1:100; Cat#66264-1-Ig, RRID: AB_2716798; Proteintech) with TDRD6 protein on adult testis sections. Consistent with our previous observation of TDRKH and TDRD6 co-staining, these newly generated data showed that TDRD6 had minimal colocalization with mitochondria in mid-pachytene spermatocytes, and primarily enriched in distinct CB precursor granules in late-pachytene and diplotene spermatocytes (revised Fig. S5a). Thus, these newly generated data support that Zenon Kit labeling works well in our system.

Fig. R1 Co-immunostaining of TDRKH and TDRD6 in adult mouse testis. Co-immunostaining of TDRKH (grayscale or red) and TDRD6 (grayscale or green) on testis sections from adult wildtype mice, with nuclei counterstained by DAPI (blue). PS, pachytene spermatocytes; DS, diplotene spermatocytes; RS, round spermatids. Scale bar, 10 μ m.

3. In general it would help the reader if instances of co-localization or absence thereof are indicated with arrows in the IF panels.

Per Reviewer's suggestion, we have indicated the instances of co-localization or absence thereof with arrows, arrowheads, or open arrowheads in all related images in revised Fig. 1 and Fig. 3.

4. Also, could the authors indicate in the manuscript which images were obtained by confocal microscopy and which one by regular microscopy? Especially for the claims of co-localization this is important.

We have now stated the imaging techniques for all related panels in revised figure legends. In particular, regular microscopy was employed for all non-colocalization staining, including the figure panels in Fig. 1c, Fig. 1d, Fig. 5d, Fig. 5e, Fig. S3a, Fig. S3b, Fig. S5c, Fig. S6a, Fig. S6c, Fig. S6d. Conversely, confocal microscopy was employed for all co-localization staining, including the figure panels in Fig. 1e, Fig. 3a, Fig. 3b, Fig. 3c, Fig. S5a.

5. An important tool is the use of an antibody, A3490, which recognizes Miwi only when it's not associated with a piRNA. Figure S4 shows the validation of this antibody. Could the authors expand this figure by providing details on the epitope it was raised

against? Also an immunofluorescence staining with this antibody on WT mouse testis material combined with 1) a marker against IMC/Mitochondria and, separately, 2) the CB (without using the Zenon kit) would be very insightful.

This antibody, now referred to as anti-MIWI^{unloaded}, is a rabbit monoclonal antibody purchased from ABclonal Technology Co., Ltd. (Catalog No.: A3490; <https://abclonal.com.cn/catalog/A3490>). This antibody was generated using a synthetic 100 aa peptide corresponding to the position 700-800 of human PIWIL1 (Q96J94). We have added the detailed information of this antibody in the **Plasmids, oligonucleotides, and antibodies** section of “Materials and Methods” in revised text (Lines 397-401). Moreover, we have further verified the specificity of this antibody using a newly generated epitope-deleted MIWI^{aa701-801} mutant (revised Fig. S4a). Please also see our response above to Reviewer 1’s Question #1.

Per Reviewer’s suggestion, we performed co-immunostaining assay of wildtype mouse testis sections using this antibody and mouse monoclonal antibodies respectively against the mitochondrial marker Cytochrome c (1:100; Cat#66264-1-Ig, RRID: AB_2716798; Proteintech; appended Fig. R2a) or the IMC & CB marker MILI (1:50, Cat#sc-377258; Santa Cruz; appended Fig. R2b). Surprisingly, we found that anti-MIWI^{unloaded} was able to detect MIWI protein in either the IMC or CB, where we assume MIWI protein is largely unloaded or loaded with piRNAs, respectively. This finding appears to be contradicted with its selective immunoprecipitation of piRNA-unloaded MIWI protein in mouse testes observed in our RIP assay (Fig. 2d). We reasoned it likely due to protein denaturation during the fixation process using 4% Paraformaldehyde (PFA) and subsequent antigen retrieval, thereby exposing the epitope for anti-MIWI^{unloaded} antibody. Indeed, we found that this antibody could not differentiate piRNA-loaded or unloaded MIWI proteins in western blotting under denaturation condition. Due to space limitation of the manuscript, we have provided the results in this response letter for inspection by the Reviewer.

Fig. R2 Immunostaining of testis sections using anti-MIWI^{unloaded} antibody. **a**, Co-immunostaining of Cytochrome c (green) and MIWI (red) on testis sections from adult wildtype mice using confocal microscopy, with nuclei counterstained by DAPI (greyscale). White arrowheads indicated colocalization sites, and white and yellow open arrowheads respectively indicated the unique localization of Cytochrome c and MIWI at non-colocalization sites. Scale bar, 10 μ m. **b**, Co-immunostaining of MILI (green) and MIWI (red) on testis sections from adult wildtype mice using confocal microscopy, with nuclei counterstained by DAPI (greyscale). White arrowheads indicated colocalization sites. Scale bar, 10 μ m.

6. Figure 3e PS/DS The TDRD6 panel suggests that loss of loading ability MIWI affects TDRD6 granules (larger granules are absent in the mutant panels). More likely the stages of these meiotic cells are not similar. Can the authors provide images of spermatocytes in a more similar stage (or alternatively expand on a necessity for MIWI loading for correct Tdrd6 localization to pre-cursor CBs)?

In general, we determined the developmental steps of meiotic cells in testis sections on the basis of morphological characteristics (DAPI staining of nuclei) and seminiferous tubule stages. In particular, we identified the Stage I-VI seminiferous tubules based on the presence of round spermatids and pachytene spermatocytes but absence of pre-leptotene spermatocytes, Stage VII-VIII seminiferous tubules based on the presence of pre-leptotene spermatocytes, Stage IX-XI seminiferous tubules based on the presence of leptotene/zygotene spermatocytes but absence of round spermatids. Indeed, we found that TDRD6 protein appeared in smaller granules in *Miwi*^{YY/YY} or *Miwi*^{YK/YK} late-pachytene and diplotene spermatocytes from Stage IX-XI seminiferous tubules (Fig. 3e). To validate this finding, we further performed γ H2AX, a spermatocyte stage marker,

and DAPI staining to determine the developmental stages of spermatocytes and spermatids. Again, we found the absence of large TDRD6 granules in *Miwi^{YY/YY}* or *Miwi^{YK/YK}* late-pachytene and diplotene spermatocytes compared to wildtype controls (revised Fig. S6c). Moreover, co-staining of TDRD6 and γ H2AX also showed smaller TDRD6 granules in *Miwi^{-/-}* late-pachytene and diplotene spermatocytes. Given that both TDRD6 and MIWI are the major components for CB precursor, we reasoned that the absence of MIWI in CB precursor might affect the size of CB precursor (Figs. 3e and S6c). Intriguingly, TDRD6 granules becomes normal in round spermatids from wildtype, *Miwi^{YY/YY}*, *Miwi^{YK/YK}* or *Miwi^{-/-}* mice. These results together suggest that the absence of MIWI in CB precursors might moderately alter the size of TDRD6 granules in late-pachytene and diplotene spermatocytes, but little affect its enrichment in the CB in round spermatids. We have described these newly generated results in the revised text (Lines 198-201).

7. Figure 3e RS, MIWI staining. Can the authors provide these images with extended overexposure? Right now I cannot determine whether signal is absent or radically reduced.

Per Reviewer's suggestion, we have overexposed the anti-MIWI staining images in RS in Fig. 3e. These overexposed images confirmed a dramatic reduction of MIWI protein in RS from *Miwi^{YY/YY}* and *Miwi^{YK/YK}* mice (appended Fig. R3). Due to space limitation of the manuscript, we have provided the results in this response letter for inspection by the Reviewer.

Fig. R3 Anti-MIWI staining images in round spermatids with extended overexposure. Overexposed co-immunostaining of MIWI (red) and TDRD6 (green) in round spermatids (RS) on testis sections from adult wildtype, *Miwi^{YY/YY}* and *Miwi^{YK/YK}* mice, with nuclei counterstained by

DAPI (blue). Scale bar, 10 μ m.

8. Figure 5c+d. The authors claim that the spermiogenic arrest in YY/YY and YK/YK mice is later than in Miwi KO. This is a very interesting observation. However, I cannot verify this with these images. Can the authors add additional data or more detailed images that underwrites their observation? Also, can the authors exclude that this difference is due to differences in genetic background of these mice (which can have a profound background on the timing of arrest in piRNA mutants)?

Per Reviewer's suggestion, we have magnified the images of ACRV1 and γ H2AX staining in revised Fig. 5d. Consistent with a previous report showing spermiogenic arrest at the beginning of round spermatid stage (Deng & Lin, *Dev Cell* 2, 819-30, 2002), round spermatids in *Miwi*^{-/-} testes exhibited proacrosome granules but all lacked acrosome caps. In contrast, the acrosome cap-containing round spermatids are readily detectable in *Miwi*^{YY/YY} and *Miwi*^{YK/YK} testes, supporting that *Miwi*^{YY/YY} and *Miwi*^{YK/YK} spermatids could progress to a later stage compared with *Miwi*^{-/-} controls. To corroborate this, we further stained testis sections using peanut agglutinin (PNA), which selectively binds to the outer acrosomal membrane in spermatids. Again, we observed the acrosome cap-containing round spermatids much more present in *Miwi*^{YY/YY} and *Miwi*^{YK/YK} testes than that in *Miwi*^{-/-} testes (revised Fig. S6d). These results together suggest that spermiogenic arrest occurs at a later stage in *Miwi*^{YY/YY} and *Miwi*^{YK/YK} mice compared to *Miwi*^{-/-} mice. We have described these data in the revised text (Lines 251-256).

In addition, we confirm that all mouse models used in this study were generated on the C57BL/6 background. Thus, we can rule out genetic background as a contributing factor to the observed phenotypic differences.

9. In the discussion there is some redundancy from line 310 on. Also, I'm curious to read the authors thoughts on Mili and the difference with Miwi in the piRNAs they generate/associate with in the light of their data. Finally, in the pull down experiment described by Vagin et al (G&D), Tdrd2/Tdrkh was indicated as an interactor of MIWI2, which could suggest that a similar mechanism is occurring in fetal germ cells too (though not with Tdrd6 since it is not expressed in these cells). What are the authors thoughts on this?

We thank the Reviewer for these insightful comments. In the discussion, we have discussed why piRNA loading plays opposite roles in regulating MIWI stability at early and late stages of spermatogenesis, which we believe it important to address the

“contradicted” findings in our present and previous studies. Please also see our response below to a related question raised by Reviewer 3.

In addition, we have added a paragraph to discuss previous findings regarding MILI and MIWI2 translocation in the revised text (Lines 321-336).

10. sentence 294 contains a linguistic error.

We thank the Reviewer for pointing out this error. We have amended “leading to its interaction with TDRKH diminished” to “leading to a diminished interaction with TDRKH” in the revised text (Lines 304-305).

11. Sentence 297, “MIWI migration” maybe “MIWI translocation”?

We have changed “MIWI migration” to “MIWI translocation” in the revised text (Line 308).

Reviewer 3 (Remarks to the Author):

Wie and colleagues set up to test if piRNA loading would impact MIWI translocation from the intermitochondrial cement (IMC), a primary piRNA processing sites to the chromatoid body (CB), a functional deployment site inside germ cells. They make use of fine, well executed mouse genetics, biochemistry, and microscopy to show that loading is important for dissociation from TDRKH (a previously known interaction and a marker of the IMC) and later association with TDRD6 (a previously known interaction and a marker of the CB). They show that arginine methylation is also somewhat dependent on loading (a previously known fact, however needed to be tested in their mutant experimental setup), and that loading is key to proper spermiogenesis (also a previously known fact, however needed to be tested in their mutant experimental setup).

Overall, the experimental system and the experiments are very neat and well executed and should be published upon revision of the current manuscript. They do in general allow the conclusions drawn in the manuscript despite not adding a lot to the field in terms of novelty. My main argument for the above statement is that, as a mechanism for dissociation of MIWI from TDRKH, “piRNA loading” is slightly beyond currents standards in the field. MIWI mutants for piRNA loading and their effect in localization and spermiogenesis have been known for about a decade. It is a high bar to ask within

the context of manuscript revisions, but the field would really benefit from understanding which allosteric changes occur upon piRNA loading and how they impact key protein-protein interactions. That would be an entire new structural biology manuscript. It does however make a point about the impact of the work presented here.

We thank the Reviewer for his/her insightful comments. We fully agree with the Reviewer's point. To date, only three PIWI-piRNA complex structures have been characterized: the crystal structures of silkworm Siwi-piRNA (Matsumoto et al., *Cell* **167**, 484-497 e489, 2016) and fly Piwi-piRNA (Yamaguchi et al., *Nat Commun* **11**, 858, 2020), and the cryo-electron microscopy structure of sponge *Ephydatia fluviatilis* Piwi (*EfPiwi*)-piRNA (Anzelon et al., *Nature* **597**, 285-289, 2021). However, none of these studies showed the structures of piRNA-unloaded PIWI proteins, preventing us from determining piRNA loading-induced allosteric changes using this existing PIWI-piRNA co-structure. Given that miRNA binding induces significant conformational changes in hAgo2 (Elkayam et al., *Cell* **150**, 100-110, 2012), we believe that PIWI proteins, belonging to Argonaute family, should similarly experience allosteric changes upon piRNA association, thereby affecting the protein-protein interactions. Indeed, we previously found that addition of a synthetic piRNA quenched >70% of the tryptophan fluorescence of wild-type MIWI but did not affect the fluorescent density of its piRNA loading-deficient Y346/347A mutant (Zhao et al., *Dev Cell* **24**, 13-25, 2013), supporting that piRNA loading induces a conformational change of MIWI. The present study showed that piRNA binding diminishes MIWI interaction with TDRKH, likely as a result of MIWI conformational alterations induced by piRNA association. Future structural and biochemical studies are poised to shed light on these complex molecular dynamics.

I present below two main and one minor points for improving the current manuscript

Role of TDRKH in occluding arginine methylation remains unclear. Is it because it mask methylation sites or is it because the same conformational changes happening upon piRNA loading exposes them? Would the same dose-dependent effect seen for TDRKH be observed for presence of piRNAs? This could be tested in the ectopic HEK cell system with/without piRNA transfections in presence/absence of the protein partners and using the A3490 antibody.

We thank the Reviewer for his/her insightful comments. With respect to how TDRKH occludes arginine methylation of MIWI protein, our working model posits that TDRKH masks its methylation sites, thereby hindering the accessibility of PRMT5 for their

methylation. Upon piRNA loading, MIWI becomes to disassociate from TDRKH and thus allows PRMT5 to methylate its arginine residues in the N-terminal domain. This hypothesis is supported by a previous study showing a robust interaction of TDRKH with the N-terminus of MIWI in an arginine methylation-independent manner (Zhang et al., *Proc Natl Acad Sci U S A* **114**, 12483-12488, 2017). Consistently, our current findings indicate that TDRKH preferentially binds to unmodified MIWI in mouse testes (Fig. 4g). More importantly, TDRKH reduces arginine methylation of MIWI in a dosage-dependent manner in co-transfected cells (Fig. 4h), directly supporting an inhibitory role of TDRKH in MIWI methylation. Taken together, our results suggest that TDRKH likely occludes the arginine methylation of MIWI protein through masking the methylation sites. We have added this point in the discussion in revised text (Lines 314-315).

Per Reviewer's suggestion, we first examined the dose-dependent effect of piRNA on MIWI methylation using the HEK293T ectopic expression system. However, we did not observe the same dose-dependent effect seen for TDRKH for presence of piRNAs (appended Fig. R4a), suggesting that piRNA loading does not directly contribute to MIWI methylation. Of note, we did not employ the A3490 (now referred to as anti-MIWI^{unloaded}) antibody in this context, as this antibody cannot differentiate piRNA-loaded and unloaded MIWI protein in western blotting under denaturation condition. Instead, we verified the piRNA-loaded state of MIWI by detecting the MIWI-associated piRNAs via autoradiography (appended Fig. R4a, panel 3). Next, we examined the effect of piRNA transfection on TDRKH-mediated inhibition of MIWI methylation. Consistent with our previous observation of an inhibitory role of TDRKH (Fig. 4h), transfection of TDRKH substantially diminished the arginine methylation of MIWI (appended Fig. R4b, lane 2), while co-transfection of piRNA partially relieved the inhibitory effect of TDRKH on MIWI methylation and simultaneously reduced MIWI interaction with TDRKH (lane 3). These results further support our working model where piRNA loading promotes MIWI disassociation with TDRKH likely through a conformational change, which in turn leads to the arginine residues in the N-terminus exposed for methylation by PRMT5. Due to space limitation of the manuscript, we have provided the results in this response letter for inspection by the Reviewer.

Fig. R4 Co-transfection of piRNA does not alter MIWI methylation but relieves the inhibitory effect of TDRKH in ectopic HEK293T cells. a, HEK293T cells were co-transfected with Flag-tagged MIWI and increasing amounts of piRNAs. Anti-Flag IP pellets were immunoblotted by the indicated antibodies. The quantity of piRNAs bound to Flag-tagged MIWI was detected through an RNA co-IP assay followed by autoradiography. Blot intensity quantification of methylated MIWI was shown in parenthesis [the one without piR-1 transfection (lane 1) is set as 1.0 after normalization with MIWI blotting]. piR-1: pUGACAUGAACACAGGUGCUCAGAUAGCUUmU. **b**, HEK293T cells were co-transfected with Flag-tagged MIWI, HA-tagged TDRKH and piRNAs. Anti-Flag IP pellets were immunoblotted by the indicated antibodies. Blot intensity quantification for methylated MIWI and TDRKH in anti-Flag IP pellets is indicated in parentheses, where lane 1 (without TDRKH and piRNAs transfection) is normalized to 1.0 based on MIWI blotting for methylated MIWI, and lane 2 (without piRNAs transfection) is similarly normalized for TDRKH.

The same corresponding author has published in the past (PMID: 23328397) data showing the importance of piRNA loading for ubiquitination and Miwi degradation. In here, mutants that fail to properly load piRNA are shown to be destabilized as proteins. Considering the contradiction, I believe the authors ought to test whether there is increased or decreased ubiquitination in piRNA loading mutants and whether the same APC/C-ubiquitination pathway is at play or if another pathway is working upon clearance of MIWI that failed on loading piRNAs.

We thank the Reviewer for his/her insightful comments. Accordingly, we have examined MIWI ubiquitination in piRNA loading-deficient *Miwi* mutant testes. Our anti-Ubiquitin (Ub) immunoblotting revealed a marked increase of MIWI ubiquitination in anti-MIWI immunoprecipitation (IP) pellets from *Miwi*^{YY/YY} and *Miwi*^{YK/YK} testes compared to wildtype control (revised Fig. S7d). These results suggest that the piRNA-binding deficient mutant MIWI is likely degraded via the ubiquitin-proteasome pathway. Given our previous study showing that the anaphase-promoting

complex/cyclosome (APC/C) mediates MIWI ubiquitination and removal in late spermatids, we next examined whether APC/C also serves as the E3 ubiquitin ligase for the degradation of piRNA-binding deficient MIWI proteins in mutant testes. Our co-IP assay revealed a significant reduction of MIWI-APC2 interaction in the late spermatid-lacking adult *Miwi*^{YY/YY} and *Miwi*^{YK/YK} testes compared to wild-type controls (appended Fig. R5), consistent with our previous study showing that the MIWI-APC/C interaction mainly occurs in late spermatids (Zhao et al., *Dev Cell* **24**, 13-25, 2013). These results suggest that MIWI ubiquitination in *Miwi*^{YY/YY} and *Miwi*^{YK/YK} testes is unlikely via the APC/C-ubiquitination pathway. It will be interesting in future studies to identify which E3 ubiquitin ligase is responsible for the ubiquitination and degradation of such piRNA-binding deficient MIWI proteins in earlier developmental stages of mouse male germ cells.

Fig. R5 MIWI little interacts with APC/C in *Miwi*^{YY/YY} and *Miwi*^{YK/YK} testes. Anti-APC2 IP pellets from adult wildtype (lane 1), *Miwi*^{YY/YY} (lane 2) and *Miwi*^{YK/YK} (lane 3) testicular lysates were immunoblotted with the indicated antibodies.

3) In addition, providing all uncropped original western blots would be ideal given the importance and extent of the biochemistry work in the manuscript. That helps other researchers better gauge and plan related future experiments.

Per Reviewer's suggestion, we have provided all uncropped original western blots in the Source Data file.

In final summary, we wish to express our gratitude to all three Reviewers for their constructive and insightful questions/suggestions. Now we have performed a number of experiments and obtained highly consistent results to address each and every concern raised. We hope you will find these changes and explanations satisfactory.

References:

- Anzelon TA, Chowdhury S, Hughes SM, Xiao Y, Lander GC, MacRae IJ. 2021. Structural basis for piRNA targeting. *Nature* 597: 285-89
- Deng W, Lin H. 2002. miwi, a murine homolog of piwi, encodes a cytoplasmic protein essential for spermatogenesis. *Dev Cell* 2: 819-30
- Ding D, Liu J, Dong K, Midic U, Hess RA, et al. 2017. PNLDC1 is essential for piRNA 3' end trimming and transposon silencing during spermatogenesis in mice. *Nat Commun* 8: 819
- Elkayam E, Kuhn CD, Tocilj A, Haase AD, Greene EM, et al. 2012. The structure of human argonaute-2 in complex with miR-20a. *Cell* 150: 100-10
- Horwich MD, Li C, Matranga C, Vagin V, Farley G, et al. 2007. The Drosophila RNA methyltransferase, DmHen1, modifies germline piRNAs and single-stranded siRNAs in RISC. *Curr Biol* 17: 1265-72
- Izumi N, Shoji K, Sakaguchi Y, Honda S, Kirino Y, et al. 2016. Identification and Functional Analysis of the Pre-piRNA 3' Trimmer in Silkworms. *Cell* 164: 962-73
- Kirino Y, Mourelatos Z. 2007. Mouse Piwi-interacting RNAs are 2'-O-methylated at their 3' termini. *Nat Struct Mol Biol* 14: 347-8
- Lim SL, Qu ZP, Kortschak RD, Lawrence DM, Geoghegan J, et al. 2015. HENMT1 and piRNA Stability Are Required for Adult Male Germ Cell Transposon Repression and to Define the Spermatogenic Program in the Mouse. *PLoS Genet* 11: e1005620
- Matsumoto N, Nishimasu H, Sakakibara K, Nishida KM, Hirano T, et al. 2016. Crystal Structure of Silkworm PIWI-Clade Argonaute Siwi Bound to piRNA. *Cell* 167: 484-97 e9
- Nishimura T, Nagamori I, Nakatani T, Izumi N, Tomari Y, et al. 2018. PNLDC1, mouse pre-piRNA Trimmer, is required for meiotic and post-meiotic male germ cell development. *EMBO Rep* 19
- Ohara T, Sakaguchi Y, Suzuki T, Ueda H, Miyauchi K, Suzuki T. 2007. The 3' termini of mouse Piwi-interacting RNAs are 2'-O-methylated. *Nat Struct Mol Biol* 14: 349-50
- Saito K, Sakaguchi Y, Suzuki T, Suzuki T, Siomi H, Siomi MC. 2007. Pimet, the Drosophila homolog of HEN1, mediates 2'-O-methylation of Piwi-interacting RNAs at their 3' ends. *Genes Dev* 21: 1603-8
- Tang W, Tu S, Lee HC, Weng Z, Mello CC. 2016. The RNase PARN-1 Trims piRNA 3' Ends to Promote Transcriptome Surveillance in *C. elegans*. *Cell* 164: 974-84
- Yamaguchi S, Oe A, Nishida KM, Yamashita K, Kajiya A, et al. 2020. Crystal structure

of *Drosophila* Piwi. *Nat Commun* 11: 858

Zhang H, Liu K, Izumi N, Huang H, Ding D, et al. 2017a. Structural basis for arginine methylation-independent recognition of PIWIL1 by TDRD2. *Proc Natl Acad Sci U S A* 114: 12483-88

Zhang Y, Guo R, Cui Y, Zhu Z, Zhang Y, et al. 2017b. An essential role for PNLDC1 in piRNA 3' end trimming and male fertility in mice. *Cell Res* 27: 1392-96

Zhao S, Gou LT, Zhang M, Zu LD, Hua MM, et al. 2013. piRNA-triggered MIWI ubiquitination and removal by APC/C in late spermatogenesis. *Dev Cell* 24: 13-25

REVIEWERS' COMMENTS

Reviewer #1 (Remarks to the Author):

I commend the authors for the excellent revision of the manuscript. The figures are beautiful and the data supportive of the conclusions.

Reviewer #2 (Remarks to the Author):

Dear authors,

all my concerns and questions have been dealt in a satisfying way.

One small remark and one suggestion:

- Fig 3b. right panel top row. White arrows are shifted in TDRKH and Merge panel.
- In some panels DAPI signal is not or hardly visible (Fig3.e bottom row). Although data is still interpretable and claims by the authors can be verified (and lack of DAPI is in this case not a big issue) my suggestion to the authors is to give DAPI imaging more thought in future work.

Reviewer #3 (Remarks to the Author):

After reading the revised manuscript and the authors' response to reviewers, it is of my opinion that most main concerns have been properly addressed or carefully considered and discussed. As a result, this is an improved version of an already good manuscript, and I support its publication without further revision. I congratulate the authors on their work.

Below is the point-by-point response to comments from the 3 reviewers.

Reviewer 1 (Remarks to the Author):

I commend the authors for the excellent revision of the manuscript. The figures are beautiful and the data supportive of the conclusions.

We sincerely thank the Reviewer for his/her high appraisal of our revision, and greatly appreciate his/her constructive suggestions for improving our manuscript during the previous round of revision.

Reviewer 2 (Remarks to the Author):

Dear authors,

all my concerns and questions have been dealt in a satisfying way.

We are delighted that the Reviewer is satisfied with our revision, and sincerely thank his/her careful inspection of our figure panels. Accordingly, we have modified the figure panel.

One small remark and one suggestion:

-Fig 3b. right panel top row. White arrows are shifted in TDRKH and Merge panel.

We thank the Reviewer for pointing out this error. We have now corrected the position of white arrows in the panel of revised Fig 3b.

-In some panels DAPI signal is not or hardly visible (Fig3.e bottom row). Although data is still interpretable and claims by the authors can be verified (and lack of DAPI is in this case not a big issue) my suggestion to the authors is to give DAPI imaging more thought in future work.

We thank the Reviewer for this suggestion and will pay more attention to DAPI staining in our future works.

Reviewer 3 (Remarks to the Author):

After reading the revised manuscript and the authors' response to reviewers, it is of my opinion that most main concerns have been properly addressed or carefully considered and discussed. As a result, this is an improved version of an already good manuscript,

and I support its publication without further revision. I congratulate the authors on their work.

We are truly grateful for the Reviewer's positive appraisal of our work and deeply appreciate his/her very insightful comments and suggestions for improving our manuscript during previous round of revision.